# Learning Weakly-supervised Contrastive Representations

**Yao-Hung Hubert Tsai**[1†], **Tianqin Li**[1†], **Weixin Liu**[1], **Peiyuan Liao**[1],
**Ruslan Salakhutdinov**[1], **Louis-Philippe Morency**[1]
[1]Carnegie Mellon University
`{yaohungt, tianqinl, weixinli, peiyuanl, rsalakhu, morency}@cs.cmu.edu`

## Abstract

We argue that a form of the valuable information provided by the auxiliary information is its implied data clustering information. For instance, considering hashtags as auxiliary information, we can hypothesize that an Instagram image will be semantically more similar with the same hashtags. With this intuition, we present a two-stage weakly-supervised contrastive learning approach. The first stage is to cluster data according to its auxiliary information. The second stage is to learn similar representations within the same cluster and dissimilar representations for data from different clusters. Our empirical experiments suggest the following three contributions. First, compared to conventional self-supervised representations, the auxiliary-information-infused representations bring the performance closer to the supervised representations, which use direct downstream labels as supervision signals. Second, our approach performs the best in most cases, when comparing our approach with other baseline representation learning methods that also leverage auxiliary data information. Third, we show that our approach also works well with unsupervised constructed clusters (e.g., no auxiliary information), resulting in a strong unsupervised representation learning approach.

## 1 Introduction

Self-supervised learning (SSL) designs learning objectives that use data's self-information but not labels. As a result, SSL empowers us to leverage a large amount of unlabeled data to learn good representations, and its applications span computer vision (Chen et al., 2020; He et al., 2020), natural language processing (Peters et al., 2018; Devlin et al., 2018) and speech processing (Schneider et al., 2019; Baevski et al., 2020). More than leveraging only data's self-information, this paper is interested in a weakly-supervised setting by assuming access to additional sources as auxiliary information for data, such as the hashtags as auxiliary attributes information for Instagram images. The auxiliary information can provide valuable but often noisy information. Hence, it raises a research challenge of how we can effectively leveraging useful information from auxiliary information.

We argue that a form of the valuable information provided by the auxiliary information is its implied data clustering information. For example, we can expect an Instagram image to be semantically more similar to the image with the same hashtags than those with different hashtags. Hence, our first step is constructing auxiliary-information-determined clusters. Specifically, we build data clusters such that the data from the same cluster have similar auxiliary information, such as having the same data auxiliary attributes. Then, our second step is to minimize the intra-cluster difference of the representations. Particularly, we present a contrastive approach - the clustering InfoNCE (Cl-InfoNCE) objective to learn similar representations for augmented variants of data within the same cluster and dissimilar representations for data from different clusters. To conclude, the presented two-stage approach leverages the structural information from the auxiliary information, then integrating the structural information into a contrastive representation learning process. See Figure 1 for an overview of our approach.

We provide the following analysis and observations to better understand our approach. First, we characterize the goodness of the Cl-InfoNCE-learned representations via the statistical relationships between the constructed clusters and the downstream labels. A resulting implication is that we can

---

[†]Equal contribution. Code available at: `https://github.com/Crazy-Jack/Cl-InfoNCE`.

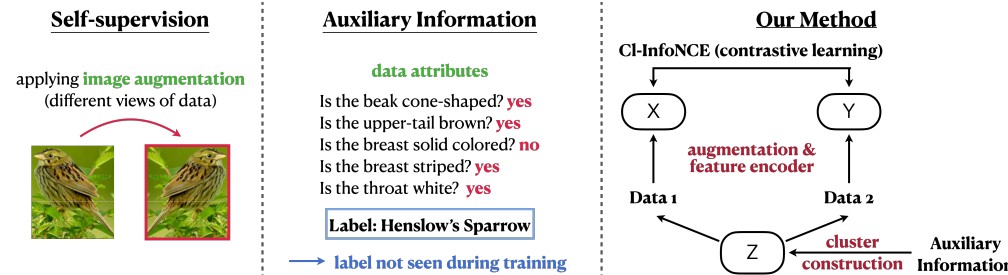

Figure 1: **Left: Self-supervision.** Self-supervised learning (SSL) uses self-supervision (the supervision from the data itself) for learning representations. An example of self-supervision is the augmented variant of the original data. **Middle: Auxiliary Information.** This paper aims to leverage auxiliary information of data for weakly-supervised representation learning. We consider data attributes (e.g., binary indicators of attributes) as auxiliary information. **Right: Our Weakly-supervised Contrastive Learning Method.** We first construct data clusters according to auxiliary information. We argue the formed clusters can provide valuable structural information of data for learning better representations. Second, we present a contrastive learning approach - the clustering InfoNCE (Cl-InfoNCE) objective to leverage the constructed clusters.

expect better downstream performance for our weakly-supervised representations when having i) higher mutual information between the labels and the auxiliary-information-determined clusters and ii) lower conditional entropy of the clusters given the labels. Second, Cl-InfoNCE generalizes recent contrastive learning objectives by changing the way to construct the clusters. In particular, when each cluster contains only one data point, Cl-InfoNCE becomes a conventional self-supervised contrastive objective (e.g., the InfoNCE objective (Oord et al., 2018)). When the clusters are built using directly the labels, Cl-InfoNCE becomes a supervised contrastive objective (e.g., the objective considered by Khosla et al. (2020)). These generalizations imply that our approach (auxiliary-information-determined clusters + Cl-InfoNCE) interpolates between conventional self-supervised and supervised representation learning.

We conduct experiments on learning visual representations using UT-zappos50K (Yu & Grauman, 2014), CUB-200-2011 (Wah et al., 2011), Wider Attribute (Li et al., 2016) and ImageNet-100 (Russakovsky et al., 2015) datasets. For the first set of experiments, we shall see how much improvement can the auxiliary information bring to us. We consider the *derivative* auxiliary information, which means the auxiliary information comes from the datasets: the discrete attributes from UT-zappos50K, CUB-200-2011, and Wider Attribute. We show that the auxiliary-information-infused weakly-supervised representations, compared to conventional self-supervised representation, have a much better performance on downstream tasks. We consider two baselines that also leverage auxiliary information: i) predicting the auxiliary-information-induced clusters with cross-entropy loss and ii) adopting the contrastive multi-view coding (CMC) (Tian et al., 2020) method when treating auxiliary information as another view of data. Our approach consistently outperforms the cross-entropy method and performs better than the CMC method in most cases. For the second set of experiments, we focus on the analysis of Cl-InfoNCE to study how well it works with unsupervised constructed clusters (K-means clusters). We find it achieves better performance comparing to the clustering-based self-supervised learning approaches, such as the Prototypical Contrastive Learning (PCL) (Li et al., 2020) method. The result suggests that the K-means method + Cl-InfoNCE can be a strong baseline for the conventional self-supervised learning setting.

## 2 RELATED WORK

**Self-supervised Learning.** Self-supervised learning (SSL) defines a pretext task as a pre-training step and uses the pre-trained features for a wide range of downstream tasks, such as object detection and segmentation in computer vision (Chen et al., 2020; He et al., 2020), question answering, and language understanding in natural language processing (Peters et al., 2018; Devlin et al., 2018) and automatic speech recognition in speech processing (Schneider et al., 2019; Baevski et al., 2020). In this paper, we focus on discussing two types of pretext tasks: clustering approaches (Caron et al., 2018; 2020) and contrastive approaches (Chen et al., 2020; He et al., 2020).

The clustering approaches jointly learn the networks' parameters and the cluster assignments of the resulting features. For example, the cluster assignments can be obtained through unsupervised clustering methods such as k-means (Caron et al., 2018), or the optimal transportation algorithms such as Sinkhorn algorithm (Caron et al., 2020). It is worth noting that the clustering approaches

enforce consistency between cluster assignments for different augmentations of the same data. The contrastive approaches learn similar representations for augmented variants of a data and dissimilar representations for different data. Examples of contrastive approaches include the InfoNCE objective (Oord et al., 2018; Chen et al., 2020; He et al., 2020), Wasserstein Predictive Coding (Ozair et al., 2019), and Relative Predictive Coding (Tsai et al., 2021a). Both the clustering and the contrastive approaches aim to learn representations that are invariant to data augmentations.

There is another line of work combining clustering and contrastive approaches, such as HUBERT (Hsu et al., 2020), Prototypical Contrastive Learning (Li et al., 2020) and Wav2Vec (Schneider et al., 2019; Baevski et al., 2020). They first construct (unsupervised) clusters from the data. Then, they perform a contrastive approach to learn similar representations for the data within the same cluster. Our approach relates to these work with two differences: 1) we construct the clusters from the auxiliary information; and 2) we present Cl-InfoNCE as a new contrastive approach and characterize the goodness for the resulting representations. Recent works like IDFD (Tao et al., 2021) aim to achieve better unsupervised clustering by using contrastive learning representations. However, Tao et al. (2021) differs from our work in that they don't directly incorporate auxiliary information into contrastive objectives.

**Weakly-supervised Learning with Auxiliary Information.** Our study relates to work on prediction using auxiliary information, by treating the auxiliary information as weak labels (Sun et al., 2017; Mahajan et al., 2018; Wen et al., 2018; Radford et al., 2021; Tan et al., 2019). The weak labels can be hashtags of Instagram images (Mahajan et al., 2018), metadata such as identity and nationality of a person (Wen et al., 2018) or corresponding textual descriptions for images (Radford et al., 2021). Compared to normal labels, the weak labels are noisy but require much less human annotation work. Surprisingly, it has been shown that the network learned with weakly supervised pre-training tasks can generalize well to various downstream tasks, including object detection and segmentation, cross-modality matching, and action recognition (Mahajan et al., 2018; Radford et al., 2021). The main difference between these works and ours is that our approach does not consider a prediction objective but a contrastive learning objective (i.e., the Cl-InfoNCE objective). An independent and concurrent work (Zheng et al., 2021) also incorporates weak labels into the contrastive learning objective. However, our method differs from Zheng et al. (2021) by the the way we construct the weak labels. We perform clustering on the annotative attributes or unsupervised k-means to obtain weak labels whereas they employ connected components labeling process. Task-wise, (Zheng et al., 2021) focuses on unsupervised (no access to data labels) and semi-supervised (access to a few data labels) representation learning, and ours focuses on weakly-supervised (access to side information such as data attributes) and unsupervised representation learning. For the common unsupervised representation learning part, we include a comparison with their method in the Appendix.

Another way to learn from auxiliary information is using multi-view contrastive coding (CMC) (Tian et al., 2020) where auxiliary information is treated as another view of the data. Specifically, CMC learns representations that can capture the joint information between the data and the accompanying auxiliary information. The main difference between CMC and our approach is that CMC leverages auxiliary information directly and Cl-InfoNCE leverages it indirectly (i.e., our approach pre-processes auxiliary information by clustering it).

## 3 METHOD

We present a two-stage approach to leverage the structural information from the auxiliary information for weakly-supervised representation learning. The first step (Section 3.1) clusters data according to auxiliary information, which we consider discrete attributes as the auxiliary information[1]. The second step (Section 3.2) presents our clustering InfoNCE (Cl-InfoNCE) objective, a contrastive-learning-based approach, to leverage the constructed clusters. We discuss the mathematical intuitions of our approach and include an information-theoretical characterization of the goodness of our learned representations. We also show that Cl-InfoNCE can specialize to recent self-supervised and supervised contrastive approaches. For notations, we use the upper case (e.g., $X$) letter to denote the random variable and the lower case (e.g., $x$) to denote the outcome from the random variable.

---

[1]Our approach generalizes to different types of auxiliary information. To help with clarity of the explanations, the paper focuses primarily on discrete attributes, but more details about other auxiliary information can be found in Appendix.

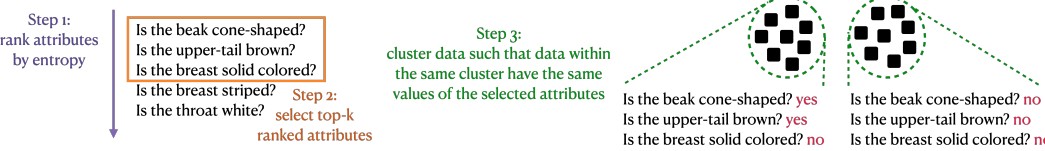

Figure 2: Cluster construction according to auxiliary information (e.g., the data attributes).

## 3.1 CLUSTER CONSTRUCTION FOR DISCRETE ATTRIBUTES

We consider discrete attributes as the auxiliary information. An example of such auxiliary information is binary indicators of attributes, such as "short/long hair", "with/without sunglasses" or "short/long sleeves", for human photos. We construct the clusters such that data within each cluster will have the same values for a set of attributes. In our running example, selecting hair and sunglasses as the set of attributes, the human photos with "long hair" and "with sunglasses" will form a cluster. Then, how we determine the set of attributes? First, we rank each attribute according to its entropy in the dataset. Note that if an attribute has high entropy, it means this attribute is distributed diversely. Then, we select the attributes with top-$k$ highest entropy, where $k$ is a hyper-parameter. The reason for this selection process is to make sure the selected attributes are informative. See Figure 2 for illustration.

## 3.2 CLUSTERING INFONCE (CL-INFONCE) OBJECTIVE

This section presents how we integrate the clustering information of data into the representation learning process. Recently, the contrastive approaches (Chen et al., 2020; Caron et al., 2020) have attracted lots of attention for self-supervised and supervised representation learning. The goal is to learn similar representations for correlated data and dissimilar representations for uncorrelated data. To be more specific, the self-supervised setting (e.g., the InfoNCE objective (Oord et al., 2018)) regards different views of the same data as correlated and distinct data as uncorrelated; the supervised setting (e.g., the supervised contrastive objective (Khosla et al., 2020)) regards the data with the same downstream label as correlated and the data with distinct labels as uncorrelated. Inspired by these methods, when performing weakly-supervised representation learning, we present to learn similar representations for data within the same cluster assignment, and vice versa. To this end, we extend from the self-supervised InfoNCE objective and introduce the clustering InfoNCE (Cl-InfoNCE) objective that takes the data clustering information into account. With the alphabets $X$ and $Y$ denoting the representations from augmented data:

$$X = \text{Feature\_Encoder}\Big(\text{Augmentation\_1}\big(\text{Data\_1}\big)\Big) \text{ and } Y = \text{Feature\_Encoder}\Big(\text{Augmentation\_2}\big(\text{Data\_2}\big)\Big)$$

and the alphabet $Z$ denoting the constructed clusters, we formulate Cl-InfoNCE as

**Definition 3.1** (Clustering-based InfoNCE (Cl-InfoNCE)).

$$\text{Cl} - \text{InfoNCE} := \sup_f \mathbb{E}_{(x_i, y_i) \sim \mathbb{E}_{z \sim P_Z}\big[P_{X|z} P_{Y|z}\big]^{\otimes n}} \left[ \frac{1}{n} \sum_{i=1}^n \log \frac{e^{f(x_i, y_i)}}{\frac{1}{n} \sum_{j=1}^n e^{f(x_i, y_j)}} \right]. \quad (1)$$

$f(x, y)$ is any function that returns a scalar from the input $(x, y)$. As suggested by prior work (Chen et al., 2020; He et al., 2020), we choose $f(x, y) = \text{cosine}\big(g(x), g(y)\big)/\tau$ to be the cosine similarity between non-linear projected $g(x)$ and $g(y)$. $g(\cdot)$ is a neural network (also known as the projection head (Chen et al., 2020; He et al., 2020)) and $\tau$ is the temperature hyper-parameter. $\{(x_i, y_i)\}_{i=1}^n$ are $n$ independent copies of $(x, y) \sim \mathbb{E}_{z \sim P_Z}\big[P_{X|z} P_{Y|z}\big]$, where it first samples a cluster $z \sim P_Z$ and then samples $(x, y)$ pair with $x \sim P_{X|z}$ and $y \sim P_{Y|z}$. Furthermore, we call $(x_i, y_i)$ as the positively-paired data ($x_i$ and $y_i$ have the same cluster assignment) and $(x_i, y_j)$ ($i \neq j$) as the negatively-paired data ($x_i$ and $y_j$ have independent cluster assignment). Note that, in practice, the expectation in equation 1 is replaced by the empirical mean of a batch of samples.

**Mathematical Intuitions.** Our objective is learning the representations $X$ and $Y$ (by updating the parameters in the Feature_Encoder) to maximize Cl-InfoNCE. At a colloquial level, the maximization pulls towards the representations of the augmented data within the same cluster and push away the representations of the augmented data from different clusters. At a information-theoretic level, we present the following:

**Theorem 3.2** (informal, Cl-InfoNCE maximization learns to include the clustering information).

$$\mathrm{Cl-InfoNCE} \leq D_{\mathrm{KL}}\left(\mathbb{E}_{P_Z}\left[P_{X|Z}P_{Y|Z}\right] \| P_X P_Y\right) \leq H(Z)$$

and the equality holds only when $H(Z|X) = H(Z|Y) = 0$,

(2)

where $H(Z)$ is the entropy of $Z$ and $H(Z|X)$ (or $H(Z|Y)$) are the conditional entropy of $Z$ given $X$ (or $Y$). Please find detailed derivations and proofs in Appendix.

The theorem suggests that Cl-InfoNCE has an upper bound $D_{\mathrm{KL}}\left(\mathbb{E}_{P_Z}\left[P_{X|Z}P_{Y|Z}\right] \| P_X P_Y\right)$, which measures the distribution divergence between the product of clustering-conditional marginal distributions (i.e., $\mathbb{E}_{P_Z}\left[P_{X|Z}P_{Y|Z}\right]$) and the product of marginal distributions (i.e., $P_X P_Y$). We give an intuition for $D_{\mathrm{KL}}\left(\mathbb{E}_{P_Z}\left[P_{X|Z}P_{Y|Z}\right] \| P_X P_Y\right)$: if $D_{\mathrm{KL}}\left(\mathbb{E}_{P_Z}\left[P_{X|Z}P_{Y|Z}\right] \| P_X P_Y\right)$ is high, then we can easily tell whether $(x, y)$ have the same cluster assignment or not. The theorem also suggests that maximizing Cl-InfoNCE results in the representations $X$ and $Y$ including the clustering information $Z$ ($\because H(Z|X) = H(Z|Y) = 0$).

**Goodness of the Learned Representations.** In Theorem 3.2, we show that maximizing Cl-InfoNCE learns the representations ($X$ and $Y$) to include the clustering ($Z$) information. Therefore, to characterize how good is the learned representations by maximizing Cl-InfoNCE or to perform cross validation, we can instead study the relations between $Z$ and the downstream labels (denoting by $T$). In particular, we can use information-theoretical metrics such as the mutual information $I(Z;T)$ and the conditional entropy $H(Z|T)$ to characterize the goodness of the learned representations. $I(Z;T)$ measures how relevant the clusters and the labels, and $H(Z|T)$ measures how much redundant information in the clusters that are irrelevant to the labels. For instance, we can expect good downstream performance for our auxiliary-information-infused representations when having high mutual information and low conditional entropy between the auxiliary-information-determined clusters and the labels. It is worth noting that, when $Z$ and $T$ are both discrete variables, computing $I(Z;T)$ and $H(Z|T)$ would be much easier than computing $I(X;T)$ and $H(X|T)$.

**Generalization of Recent Self-supervised and Supervised Contrastive Approaches.** Cl-InfoNCE (equation 1) serves as an objective that generalizes to different levels of supervision according to how we construct the clusters ($Z$). When $Z = $ instance id (i.e., each cluster only contains one instance), $\mathbb{E}_{P_Z}\left[P_{X|Z}P_{Y|Z}\right]$ specializes to $P_{XY}$ and Cl-InfoNCE specializes to the InfoNCE objective (Oord et al., 2018), which aims to learn similar representations for augmented variants of the same data and dissimilar representations for different data. InfoNCE is the most popular used self-supervised contrastive learning objective (Chen et al., 2020; He et al., 2020; Tsai et al., 2021b). When $Z = $ downstream labels, Cl-InfoNCE specializes to the objective described in *Supervised Contrastive Learning* (Khosla et al., 2020), which aims to learn similar representations for data that are from the same downstream labels and vice versa. In our paper, the clusters $Z$ are determined by the auxiliary information, and we aim to learn similar representations for data sharing the same auxiliary information and vice versa. This process can be understood as weakly supervised contrastive learning. To conclude, Cl-InfoNCE is a clustering-based contrastive learning objective. By differing its cluster construction, Cl-InfoNCE interpolates among unsupervised, weakly supervised, and supervised representation learning.

# 4 EXPERIMENTS

We given an overview of our experimental section. Section 4.1 discusses the datasets. We consider discrete attribute information as auxiliary information for data. Next, in Section 4.2, we explain the methodology that will be used in the experiments. Section 4.3 presents the first set of the experiments, under a weakly-supervised setting, to manifest the effectiveness of our approach and the benefits of taking the auxiliary information into account. Last, to study the effect of Cl-InfoNCE alone, Section 4.4 presents the second set of the experiments under a unsupervised setting. We also conduct comparison experiments with another independent and concurrent weakly supervised contrastive learning work (Zheng et al., 2021) in the Appendix.

## 4.1 DATASETS

We consider the following datasets. **UT-zappos50K** (Yu & Grauman, 2014): It contains $50,025$ shoes images along with 7 discrete attributes as auxiliary information. Each attribute follows a binomial

distribution, and we convert each attribute into a set of Bernoulli attributes, resulting in a total of 126 binary attributes. There are 21 shoe categories. **Wider Attribute** (Li et al., 2016): It contains 13, 789 images, and there are several bounding boxes in an image. The attributes are annotated per bounding box. We perform OR operation on attributes from different bounding boxes in an image, resulting in 14 binary attributes per image as the auxiliary information. There are 30 scene categories. **CUB-200-2011** (Wah et al., 2011): It contains 11, 788 bird images with 200 binary attributes as the auxiliary information. There are 200 bird species. For the second set of the experiments, we further consider the **ImageNet-100** (Russakovsky et al., 2015) dataset. It is a subset of the ImageNet-1k object recognition dataset (Russakovsky et al., 2015), where we select 100 categories out of $1, 000$, resulting in around 0.12 million images.

## 4.2 Methodology

Following Chen et al. (2020), we conduct experiments on pre-training visual representations and then evaluating the learned representations using the linear evaluation protocol. In other words, after the pre-training stage, we fix the pre-trained feature encoder and then categorize test images by linear classification results. We select ResNet-50 (He et al., 2016) as our feature encoder across all settings. Note that our goal is learning representations (i.e, $X$ and $Y$) for maximizing the Cl-InfoNCE objective (equation equation 1). Within Cl-InfoNCE, the positively-paired representations $(x, y^+) \sim \mathbb{E}_{z \sim P_Z}[P_{X|z}P_{Y|z}]$ are the learned representations from augmented images from the same cluster $z \sim P_Z$ and the negatively-paired representations $(x, y^-) \sim P_X P_Y$ are the representations from arbitrary two images. We leave the network designs, the optimizer choices, and more details for the datasets in Appendix.

Before delving into the experiments, we like to recall that, in Section 3.2, we discussed using the mutual information $I(Z;T)$ and the conditional entropy $H(Z|T)$ between the clusters ($Z$) and the labels ($T$) to characterize the goodness of Cl-InfoNCE's learned representations. To prove this concept, on UT-Zappos50K, we synthetically construct clusters for various $I(Z;T)$ and $H(Z|T)$ followed by applying Cl-InfoNCE. We present the results in the right figure. Our empirical results are in accordance with the statements that the clusters with higher $I(Z;T)$ and lower $H(Z|T)$ will lead to higher downstream performance. In later experiments, we will also discuss these two information-theoretical metrics.

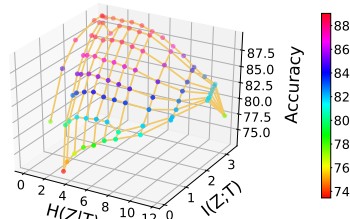

Figure 3: $I(Z;T)$ represents how relevant the clusters and the labels; higher is better. $H(Z|T)$ represents the redundant information in the clusters for the labels; lower is better.

## 4.3 Experiment I: Auxiliary-Information-Determined Clusters + Cl-InfoNCE

We like to understand how well Cl-InfoNCE can be combined with the auxiliary information. For this purpose, we select the data discrete attributes as the auxiliary information, construct the clusters ($Z$) using the discrete attributes (see Section 3.1 and Figure 2), and then adopt attributes-determined clusters for Cl-InfoNCE. Recall our construction of data-attributes-determined clusters: we select the attributes with top-$k$ highest entropy and then construct the clusters such that the data within a cluster will have the same values over the selected attributes. $k$ is the hyper-parameter. Note that our method considers a weakly supervised setting since the data attributes can be seen as the data's weak supervision.

We dissect the experiments into three parts. First, we like to study the effect of the hyper-parameter $k$ and select its optimal value. Note that different choices of $k$ result in different constructed clusters $Z$. Our study is based on the information-theoretical metrics (i.e., $I(Z;T)$ and $H(Z|T)$ between the constructed clusters ($Z$) and the labels ($T$)) and their relations with the downstream performance of the learned representations. Second, we perform comparisons between different levels of supervision. In particular, we include the comparisons with the supervised ($Z =$ downstream labels $T$) and the conventional self-supervised ($Z =$ instance ID) setting for our method. We show in Section 3.2, the supervised setting is equivalent to the Supervised Contrastive Learning objective (Khosla et al., 2020) and the conventional self-supervised setting is equivalent to SimCLR (Chen et al., 2020). Third, we include baselines that leverage the auxiliary information: i) learning to predict the clusters assignments using cross-entropy loss and ii) treating auxiliary information as another view of data when using the contrastive multi-view coding (CMC) (Tian et al., 2020).

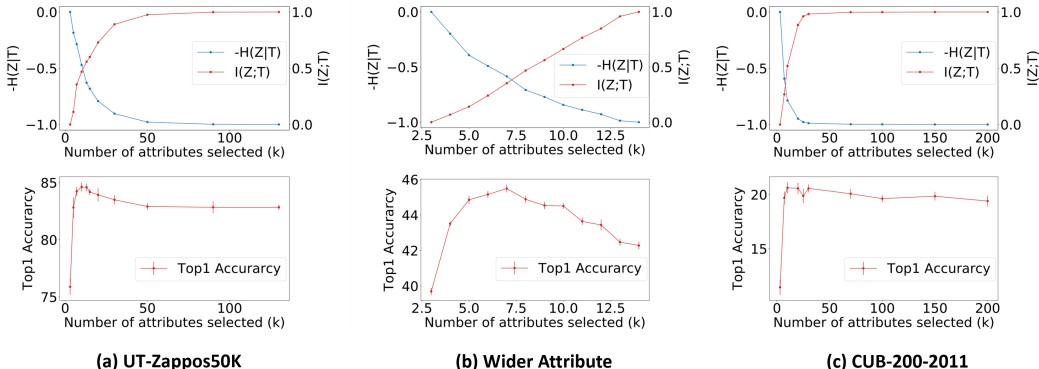

| | | | |
|---|:---:|:---:|:---:|
| **(a) UT-Zappos50K** | **(b) Wider Attribute** | **(c) CUB-200-2011** | |

Figure 4: Experimental results for attributes-determined clusters + Cl-InfoNCE by tuning the hyper-parameter $k$ when constructing the clusters. Note that we select attributes with top-$k$ highest entropy, and we construct the clusters such that the data within a cluster would have the same values for the selected attributes. $Z$ are the constructed clusters, and $T$ are the downstream labels. We find the intersection between the re-scaled $I(Z;T)$ and the re-scaled $-H(Z|T)$ gives us the best downstream performance.

| Method | UT-Zappos50K | | Wider Attribute | | CUB-200-2011 | |
|---|:---:|:---:|:---:|:---:|:---:|:---:|
| | Top-1 Acc. | Top-5 Acc. | Top-1 Acc. | Top-5 Acc. | Top-1 Acc. | Top-5 Acc. |
| *(Supervised) Labels + Cl-InfoNCE* | | | | | | |
| SupCon (Khosla et al., 2020) | 89.0±0.4 | 99.4± 0.3 | 49.9±0.8 | 76.2±0.2 | 59.9±0.7 | 78.8± 0.3 |
| *(Weakly-supervised) Attributes-Determined Clusters + Cl-InfoNCE* | | | | | | |
| Ours | 84.6±0.4 | 99.1±0.2 | 45.5±0.2 | 75.4±0.2 | 20.6± 0.5 | 47.0±0.5 |
| *(Self-supervised) Instance-ID + Cl-InfoNCE* | | | | | | |
| SimCLR (Chen et al., 2020) | 77.8±1.5 | 97.9±0.8 | 40.2±0.9 | 73.0±0.3 | 14.1± 0.7 | 35.2±0.6 |

Table 1: Experimental results for different supervision levels with the presented Cl-InfoNCE objective. For weakly-supervised methods that consider attributes-determined clusters, we report the best results by tuning the hyper-parameter $k$. The results suggest that, with the help of auxiliary information, we can better close the performance gap between supervised and self-supervised representations.

**Part I - Effect of the hyper-parameter $k$.** To better understand the effect of the hyper-parameter $k$ for constructing the attributes-determined clusters, we study the information-theoretical metrics between $Z$ and $T$ and report in Figure 4. Note that, to ensure the same scales for $I(Z;T)$ and $H(Z|T)$ across different datasets, we normalize $I(Z;T)$ and $H(Z|T)$ using

$$I(Z;T) \leftarrow \frac{I(Z;T) - \min_Z I(Z;T)}{\max_Z I(Z;T) - \min_Z I(Z;T)} \quad \text{and} \quad H(Z|T) \leftarrow \frac{H(Z|T)) - \min_Z H(Z|T)}{\max_Z H(Z|T) - \min_Z H(Z|T)}.$$

As $k$ increases, the mutual information $I(Z;T)$ increases but the conditional entropy $H(Z|T)$ also increases. Hence, although considering more attributes leads to the clusters that are more correlated to the downstream labels, the clusters may also contain more downstream-irrelevant information. This is in accord with our second observation that, as $k$ increases, the downstream performance first increases then decreases. Therefore, we only need a partial set of the most informative attributes (those with high entropy) to determine the clusters. Next, we observe that the best performing clusters happen at the intersection between $I(Z;T)$ and negative $H(Z|T)$. This observation helps us study the trade-off between $I(Z;T)$ and $H(Z|T)$ and suggests an empirical way to select the optimal $k$ that achieves the best performance. It is worth noting that the above process of determining the optimal $k$ does not require directly evaluating the learned representations.

**Part II - Interpolation between Different Supervision Levels.** In Section 3.2, we discussed that, by altering the designs of the clusters, our presented approach specializes to the conventional self-supervised contrastive method - SimCLR (Oord et al., 2018) and the supervised contrastive method - SupCon (Khosla et al., 2020). In particular, our approach specializes to SimCLR when considering augmented variants of each instance as a cluster and specializes to SupCon when considering instances with the same downstream label as a cluster. Hence, we can interpolate different supervision levels of our approach and study how auxiliary information of data can help improve representation learning.

| Method | UT-Zappos50K | | Wider Attribute | | CUB-200-2011 | |
|---|---|---|---|---|---|---|
| | Top-1 Acc. | Top-5 Acc. | Top-1 Acc. | Top-5 Acc. | Top-1 Acc. | Top-5 Acc. |
| *Self-supervised Representation Learning* | | | | | | |
| MoCo (He et al., 2020) | 83.4±0.2 | 99.1±0.3 | 41.0±0.7 | 74.0±0.4 | 13.8±0.7 | 36.5±0.5 |
| SimCLR (Chen et al., 2020) | 77.8±1.5 | 97.9±0.8 | 40.2±0.9 | 73.0±0.3 | 14.1± 0.7 | 35.2±0.6 |
| *Weakly-supervised Representation Learning* | | | | | | |
| Contrastive Multi-view Coding (CMC) (Tian et al., 2020) | 83.7±0.5 | **99.2±0.3** | 34.1±1.2 | 65.3±0.7 | **32.7±0.3** | **61.8±0.5** |
| Attributes-Determined Clusters + Cross-Entropy Loss | 82.7±0.7 | 99.0±0.3 | 39.4±0.6 | 68.6±0.2 | 17.5±1.0 | 46.0±0.8 |
| Attributes-Determined Clusters + Cl-InfoNCE (Ours) | **84.6±0.4** | 99.1±0.2 | **45.5±0.2** | **75.4±0.2** | 20.6± 0.5 | 47.0±0.5 |

Table 2: Experimental results for weakly-supervised representation methods that leverage auxiliary information and self-supervised representation methods. Best results are highlighted in bold. The results suggest that our method outperforms the weakly-supervised baselines in most cases with the exception that the CMC method performs better than our method on the CUB-200-2011 dataset.

We present the results in Table 1 with different cluster constructions along with Cl-InfoNCE. We use the top-1 accuracy on Wider Attribute for discussions. We find the performance grows from low to high when having the clusters as instance ID (40.2), attributes-determined clusters (45.5) to labels (49.9). This result suggests that CL-InfoNCE can better bridge the gap with the supervised learned representations by using auxiliary information.

**Part III - Comparisons with Baselines that Leverage Auxiliary Information.** In the last part, we see that Cl-InfoNCE can leverage auxiliary information to achieve a closer performance to supervised representations than self-supervised representations. Nonetheless, two questions still remain: 1) is there another way to leverage auxiliary information other than our method (attributes-determined clusters + Cl-InfoNCE), and 2) is the weakly-supervised methods (that leverages auxiliary information) always better than self-supervised methods? To answer these two questions, in Table 2, we include the comparisons among weakly-supervised representation learning baselines that leverage auxiliary information (Attributes-determined clusters + cross-entropy loss and Contrastive Multi-view Coding (CMC) (Tian et al., 2020) when treating auxiliary information as another view of data) and self-supervised baselines (SimCLR (Oord et al., 2018) and MoCo (He et al., 2020)).

First, we find that using auxiliary information does not always guarantee better performance than not using it. For instance, for top-1 acc. on Wider Attribute dataset, predicting the attributes-determined clusters using the cross-entropy loss (39.4) or treating auxiliary information as another view of data then using CMC (34.1) perform worse than the SimCLR method (40.2), which does not utilize the auxiliary information. The result suggests that, although auxiliary information can provide useful information, how we can effectively leverage the auxiliary information is even more crucial.

Second, we observe that our method constantly outperforms the baseline - Attributes-Determined Clusters + Cross-Entropy loss. For instance, on ZT-Zappos50K, our method achieves 84.6 top-1 accuracy while the baseline achieves 82.7 top-1 accuracy. Note that both our method and the baseline consider constructing clusters according to auxiliary information. The difference is that our method adopts the contrastive approach - Cl-InfoNCE, and the baseline considers to adopt cross-entropy loss on an additional classifier between the representations and the clusters. Our observation is in accordance with the observation from a prior work (Khosla et al., 2020). It shows that, compared to the cross-entropy loss, the contrastive objective (e.g., our presented Cl-InfoNCE) is more robust to natural corruptions of data, stable to hyper-parameters and optimizers settings, and enjoying better performance.

Last, we compare our method with the CMC method. We see that although our method performs better on UT-zappos50K (84.6 over 83.7) and Wider Attributes (45.5 over 34.1) dataset, CMC achieves significantly better results on CUB-200-2011 (32.7 over 20.6) dataset. To explain such differences, we recall that 1) the CMC method leverages the auxiliary information directly, while our method leverages the auxiliary information indirectly (we use the structural information implied from the auxiliary information); and 2) the auxiliary information used in UT-zappos50K and Wider Attributes contains relatively little information (i.e., consisting of less than 20 discrete attributes), and the auxiliary information used in CUB-200-2011 contains much more information (i.e., consisting of 312 discrete attributes). We argue that since CMC leverages the auxiliary information directly, it shall perform better with more informative auxiliary information. On the other hand, Cl-InfoNCE performs better with less informative auxiliary information.

| Method | UT-Zappos50K
Top-1 (Accuracy) | Wider Attribute
Top-1 (Accuracy) | CUB-200-2011
Top-1 (Accuracy) | ImageNet-100
Top-1 (Accuracy) |
|---|---|---|---|---|
| *Non-clustering-based Self-supervised Approaches* | | | | |
| SimCLR (Chen et al., 2020) | 77.8±1.5 | 40.2±0.9 | 14.1±0.7 | 58.2±1.7 |
| MoCo (He et al., 2020) | 83.4±0.5 | 41.0±0.7 | 13.8±0.5 | 59.4±1.6 |
| *Clustering-based Self-supervised Approaches (# of clusters = 1K/ 1K/ 1K/ 2.5K)* | | | | |
| PCL (Li et al., 2020) | 82.4±0.5 | 41.0±0.4 | 14.4±0.5 | 68.9±0.7 |
| K-means + Cl-InfoNCE (ours) | **84.5±0.4** | **43.6±0.4** | **17.6±0.2** | **77.9±0.7** |

Figure 5: Experimental results under conventional self-supervised setting (pre-training using no label supervision and no auxiliary information). **Left:** We compare our method (K-means clusters + Cl-InfoNCE) with self-supervised approaches that leverage and do not consider unsupervised clustering. The downstream performance is reported using the linear evaluation protocal (Chen et al., 2020). **Right:** For our method and Prototypical Contrastive Learning (PCL), we plot the mutual information ($I(Z; T)$) and the conditional entropy ($H(Z|T)$) versus training epochs. $Z$ are the unsupervised clusters, and $T$ are the downstream labels. The number of clusters is determined via grid search over $\{500, 1, 000, 5, 000, 10, 000\}$.

## 4.4 EXPERIMENT II: K-MEANS CLUSTERS + CL-INFONCE

So far, we see how we can combine auxiliary-information-determined clusters and Cl-InfoNCE to learn good weakly-supervised representations. Now, we would like to show that Cl-InfoNCE can also learn good self-supervised representations without auxiliary information. To this end, we construct unsupervised clusters (e.g., k-means clusters on top of the learned representations) for Cl-InfoNCE. Similar to the EM algorithm, we iteratively perform the k-means clustering to determine the clusters for the representations, and then we adopt Cl-InfoNCE to leverage the k-means clusters to update the representations. We select thet Prototypical Contrastive Learning (PCL) (Li et al., 2020) as the baseline of the clustering-based self-supervised approach. In particular, PCL performs data log-likelihood maximization by assuming data are generated from isotropic Gaussians. It considers the MLE objective, where the author makes a connection with contrastive approaches (Chen et al., 2020; He et al., 2020). The clusters in PCL are determined via MAP estimation. For the sake of the completeness of the experiments, we also include the non-clustering-based self-supervised approaches, including SimCLR (Chen et al., 2020) and MoCo (He et al., 2020). Note that this set of experiments considers the conventional self-supervised setting, in which we can leverage the information neither from labels nor from auxiliary information.

**Results.**    We first look at the left table in Figure 5. We observe that, except for ImageNet-100, there is no obvious performance difference between the non-clustering-based (i.e., SimCLR and MoCo) and the clustering-based baseline (i.e., PCL). Since ImageNet-100 is a more complex dataset comparing to the other three datasets, we argue that, when performing self-supervised learning, discovering latent structures in data (via unsupervised clustering) may best benefit larger-sized datasets. Additionally, among all the approaches, our method reaches the best performance. The result suggests our method can be as competitive as other conventional self-supervised approaches.

Next, we look at the right plot in Figure 5. We study the mutual information $I(Z; T)$ and the conditional entropy $H(Z|T)$ between the unsupervised constructed clusters $Z$ and the downstream labels $T$. We select our method and PCL, providing the plot of the two information-theoretical metrics versus the training epoch. We find that, as the number of training epochs increases, both methods can construct unsupervised clusters that are more relevant (higher $I(Z; T)$) and contain less redundant information (lower $H(Z|T)$) about the downstream label. This result suggests that the clustering-based self-supervised approaches are discovering the latent structures that are more useful for the downstream tasks. It is worth noting that our method consistently has higher $I(Z; T)$ and lower $H(Z|T)$ comparing to PCL.

## 5 CONCLUSION AND DISCUSSIONS

In this paper, we introduce the clustering InfoNCE (Cl-InfoNCE) objective that leverages the implied data clustering information from auxiliary information or data itself for learning weakly-supervised representations. Our method effectively brings the performance closer to the supervised learned representations compared to the conventional self-supervised learning approaches, therefore improving pretraining quality when limited information is at hand. In terms of limitation, our approach requires clustering based on auxiliary information or data itself. This process sometimes could pose additional computational cost. In addition, clustering on auxiliary information or data will also lose precision. Tackling these problems would be our further research direction.

## ETHICS STATEMENT

All authors of this work have read the ICLR code of ethics and commit to adhering to it. There is no ethical concern in this work.

## REPRODUCIBILITY STATEMENT

The code for reproducing our results in the experiment section can be found at https://github.com/Crazy-Jack/Cl-InfoNCE.

## ACKNOWLEDGEMENTS

The authors would like to thank the anonymous reviewers for helpful comments and suggestions. This work is partially supported by the National Science Foundation IIS1763562, IARPA D17PC00340, ONR Grant N000141812861, Facebook PhD Fellowship, BMW, National Science Foundation awards 1722822 and 1750439, and National Institutes of Health awards R01MH125740, R01MH096951 and U01MH116925. Any opinions, findings, conclusions, or recommendations expressed in this material are those of the author(s) and do not necessarily reflect the views of the sponsors, and no official endorsement should be inferred.

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

## APPENDIX A    COMPARISON WITH OTHERS RELATED IN LITERATURE

### A.1    EXPERIMENTS: COMPARISON WITH WEAKLY SUPERVISED CONTRASTIVE LEARNING(ZHENG ET AL., 2021)

We would like to point out that a concurrent work Zheng et al. (2021) presented a similar idea on weakly-supervised contrastive learning in ICCV 2021. We would like to point out the reason it is a concurrent work with ours. Zheng et al. (2021) is made publicly available on 10/05/2021, which is the same day as the the paper submission deadline for ICLR'22. To be more precise, ICCV publicly released this paper on 10/05/2021, and the paper's arxiv version and code are available on 10/10/2021. The zero time overlap suggests that our two works are independent and concurrent.

**Similarity and Difference**    We acknowledge that the two works share the similar idea of utilizing weak labels of data in contrastive learning. Zheng et al. (2021) motivates by preventing class collision during instance-wise contrastive learning (random data that belongs to the same category will possibly get falsely pushed away in instance-wise contrastive learning), and ours motivates by exploring the structural information of data within contrastive learning, followed by providing information-theoretic analysis to explain how different structural information can affect the learned representations. Task-wise Zheng et al. (2021) focuses on unsupervised (no access to data labels) and semi-supervised (access to a few data labels) representation learning, and ours focuses on weakly supervised (access to side information such as data attributes) and unsupervised representation learning. For the common unsupervised representation learning part, Zheng et al. (2021) presents to generate weak labels using connected components labeling process, and ours generates weak labels using K-means clustering.

**Empirical Results**    We observed that the performance on ImageNet-100 reported in [1] looks better than ours (79.77 Zheng et al. (2021) v.s. ours 77.9 Figure 5). However, the experimental settings differ a lot. First, the **datasets** are different despite the same name: Zheng et al. (2021) considers ImageNet-100 by selecting the first 100 class of the ILSVRC 2012 challenge, and we select a different set of 100 classes (details shown in the Appendix E). Second, the **batch size** is different: Zheng et al. (2021) considers 2048 , and ours considers 128. Third, the **projection heads in architecture** are different: Zheng et al. (2021) uses 2 projection heads (each with 4096 hidden units) with two objectives, one is for InfoNCE and the other is for the proposed Weakly Supervised Contrastive Learning loss; whereas ours uses one projection head with 2048 hidden units for Cl-InfoNCE objective only. Although our main experiments have demonstrated that Cl-InfoNCE alone can achieve competitive performance, we acknowledge that adding InfoNCE objective with an additional linear projection head would further improve the learned representation.

To fairly compare our Cl-InfoNCE loss with their proposed Weakly Supervised Contrastive objective, we add an additional head trained with InfoNCE along with our Cl-InfoNCE objective. Experiments are conducted on our version of ImageNet100 with the controlled set up: same network architecture of resnet50, same batch size of 384, same training epochs of 200, same projection head (2048-2048-128), the same optimizer and linear evaluation protocols, etc. Our Kmeans cluster number K is chosen to be 2500 via a grid search from $\{100, 1000, 2500, 5000, 10,000\}$. The results are shown below Table 3.

| Method | Top-1 Accuracy (%) |
|---|---|
| Weakly Supervised Contrastive Learning Zheng et al. (2021) | $82.2 \pm 0.4$ |
| CL-InfoNCE + Kmeans (Ours) | $82.6 \pm 0.2$ |

Table 3: Results on ImageNet-100 (Russakovsky et al., 2015) compare with a concurrent and independent work Zheng et al. (2021).

From the results, we can see that the two methods' performances are similar. Our work and theirs [1] are done independently and concurrently, and both works allow a broader understanding of weakly supervised contrastive learning.

### A.2 EXPERIMENTS: COMPARISON WITH IDFD (TAO ET AL., 2021)

IDFD (Tao et al., 2021) presents to learn representations that are clustering friendly (from a spectral clustering viewpoint) during the instance discrimination (ID) contrastive learning process. Although it includes both ideas of clustering and contrastive learning, IDFD (Tao et al., 2021) differs from our paper fundementally because they does not utilize the constructed clusters as weak labels to train contrastive objective. However, IDFD (Tao et al., 2021) can still be considered as a self-supervised representation learning method, hence we perform experiments to compare our unsupervised setting (Cl-InfoNCE + Kmeans method) with their proposed IDFD on CIFAR10 Dataset (Krizhevsky et al., 2009). To provide a fair comparison with IDFD (Tao et al., 2021), we stick to the training paradigm of IDFD where they replaces Resnet-50 with Resnet-18. The batch size of 128 is used following their report. Since IDFD (Tao et al., 2021) was focusing on clustering quality and didn't report the linear evaluation protocol, we use the released code of IDFD (Tao et al., 2021) to re-train the model meanwhile using both the cluster accuracy and the linear evaluation protocal as evaluation metrics. We train both methods for 1000 epochs for a fair comparison. The results are presented in Table 4.

| Methods | 550 Epochs | | 750 Epochs | | 1000 Epochs | |
|---|---|---|---|---|---|---|
| | Linear Eva. | Clustering Acc. | Linear Eva. | Clustering Acc. | Linear Eva. | Clustering Acc. |
| IDFD (Tao et al., 2021) | 81.7 ± 0.9 | 66.9 ± 1.4 | 82.1 ± 1.2 | 70.1 ± 1.7 | 84.4 ± 1.1 | 77.7 ± 1.5 |
| Cl-InfoNCE + Kmeans (ours) | **89.5 ± 0.6** | **78.3 ± 1.9** | **90.2 ± 0.7** | **79.1 ± 2.1** | **90.7 ± 0.8** | **81.1 ± 1.2** |

Table 4: Comparison with IDFD (Tao et al., 2021) on CIFAR10 dataset (Krizhevsky et al., 2009). Two evaluation metrics, Linear evaluation and clustering accuracy are measured during the training epochs. The Kmeans hyperparameter K is determined followed by a grid search from {10, 100, 1000, 2500}.

Note that (Tao et al., 2021) proposed 2 methods (IDFD and IDFO), we choose the compare with IDFD because **(i)** IDFO is very unstable, **(ii)** IDFD/IDFO perform at-par for the best performance based on Figure2 in (Tao et al., 2021) and **(iii)** (Tao et al., 2021) only officially releases code for IDFD. We can observe that our method exceeds IDFD on in terms of top-1 classification accuracy during linear evaluation and also improve the raw clustering accuracy score, indicating integrating weak labels from unsupervised clustering with contrastive objectives would help both representation learning and the unsupervised clustering task.

## APPENDIX B    DATA'S HIERARCHY INFORMATION AS AUXILIARY INFORMATION

In the main text, we select the discrete attributes as the auxiliary information of data, then presenting data cluster construction according to the discrete attributes. We combine the constructed clusters and the presented Cl-InfoNCE objective together for learning weakly-supervised representations. In this section, we study an alternative type of the auxiliary information - data labels' hierarchy information, more specifically, the WordNet hierarchy (Miller, 1995), illustrated in the right figure. In the example, we present the WordNet hierarchy of the label "Henslow's Sparrow", where only the WordNet hierarchy would be seen during training but not the label.

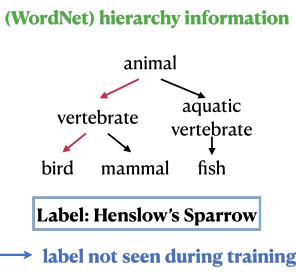

### B.1    CLUSTER CONSTRUCTION FOR WORDNET HIERARCHY

How do we construct the data clusters according to the WordNet hierarchy? In the above example, "vertebrate" and "bird" can be seen as the coarse labels of data. We then construct the clusters such that data within each cluster will have the same coarse label. Now, we explain how we determine which coarse labels for the data. First, we represent the Word-Net hierarchy into a tree structure (each children node has only one parent node). Then, we choose the coarse labels to be the nodes in the level $l$ in the WordNet tree hierarchy (the

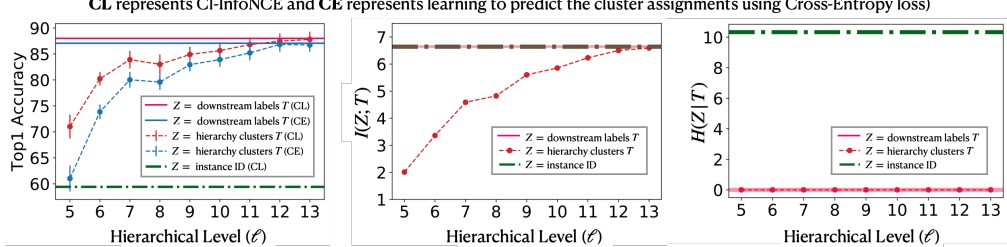

Figure 6: Experimental results on ImageNet-100 for Cl-InfoNCE under supervised (clusters $Z =$ downstream labels $T$), weakly supervised ($Z =$ hierarchy clusters) and conventional self-supervised ($Z =$ instance ID) setting. We also consider the baseline - learning to predict the clustering assignment using the cross-entropy loss. Note that we construct the clusters such that the data within a cluster have the same parent node in the level $\ell$ in the data's WordNet tree hierarchy. Under this construction, the root node is of the level 1, and the downstream labels are of the level 14. $I(Z;T)$ is the mutual information, and $H(Z|T)$ is the conditional entropy.

root node is level 1). $l$ is a hyper-parameter. We illustrate the process in the below figure.

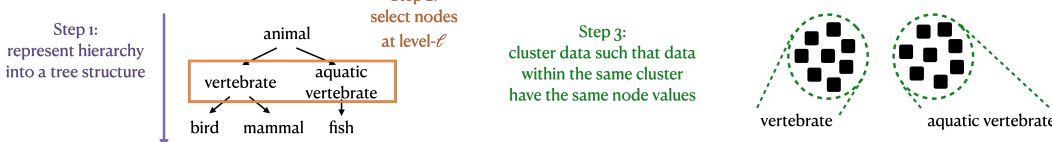

## B.2 Experiments: Data-Hierarchy-Determined Clusters + Cl-InfoNCE

The experimental setup and the comparing baselines are similar to Section 4.3 in the main text, but now we consider the WordNet (Miller, 1995) hierarchy as the auxiliary information. As discussed in prior subsection, we construct the clusters $Z$ such that the data within a cluster have the same parent node in the level $l$ in the data's WordNet tree hierarchy. $l$ is the hyper-parameter[2].

**Results.** Figure 6 presents our results. First, we look at the leftmost plot, and we have several similar observations when having the data attributes as the auxiliary information. One of them is that our approach consistently outperforms the auxiliary-information-determined clusters + cross-entropy loss. Another of them is that the weakly supervised representations better close the gap with the supervised representations. Second, as discussed in prior subsection, the WordNet data hierarchy clusters can be regarded as the coarse labels of the data. Hence, when increasing the hierarchy level $l$, we can observe the performance improvement (see the leftmost plot) and the increasing mutual information $I(Z;T)$ (see the middle plot) between the clusters $Z$ and the labels $T$. Note that $H(Z|T)$ remains zero (see the rightmost plot) since the coarse labels (the intermediate nodes) can be determined by the downstream labels (the leaf nodes) under the tree hierarchy structure. Third, we discuss the conventional self-supervised setting with the special case when $Z =$ instanced ID. $Z$ as the instance ID has the highest $I(Z;T)$ (see the middle plot) but also the highest $H(Z|T)$ (see the rightmost plot). And we observe that the conventional self-supervised representations perform the worse (see the leftmost plot). We conclude that, when using clustering-based representation learning approaches, we shall not rely purely on the mutual information between the data clusters and the downstream labels to determine the goodness of the learned representations. We shall also take the redundant information in the clusters into account.

## Appendix C   Theoretical Analysis

In this section, we provide theoretical analysis on the presented Cl-InfoNCE objective. We recall the definition of Cl-InfoNCE and our presented theorem:

**Definition C.1** (Clustering-based InfoNCE (Cl-InfoNCE), restating Definition 3.1 in the main text)**.**

$$\mathrm{Cl} - \mathrm{InfoNCE} := \sup_{f} \mathbb{E}_{(x_i,y_i)\sim\mathbb{E}_{z\sim P_Z}\left[P_{X|z}P_{Y|z}\right]^{\otimes n}}\left[\frac{1}{n}\sum_{i=1}^{n}\log\frac{e^{f(x_i,y_i)}}{\frac{1}{n}\sum_{j=1}^{n}e^{f(x_i,y_j)}}\right],$$

---

[2]Note that we do not compare with the CMC method for fair comparisons with other method. The reason is that the CMC method will leverage the entire tree hierarchy, instead of a certain level in the tree hierarchy.

**Theorem C.2** (informal, Cl-InfoNCE maximization learns to include the clustering information, restating Theorem 3.2 in the main text)**.**

$$\mathrm{Cl} - \mathrm{InfoNCE} \leq D_{\mathrm{KL}}\left(\mathbb{E}_{P_Z}\left[P_{X|Z}P_{Y|Z}\right] \| P_X P_Y\right) \leq H(Z)$$
$$\text{and the equality holds only when } H(Z|X) = H(Z|Y) = 0.$$

Our goal is to prove Theorem C.2. For a better presentation flow, we split the proof into three parts:

- Proving $\mathrm{Cl} - \mathrm{InfoNCE} \leq D_{\mathrm{KL}}\left(\mathbb{E}_{P_Z}\left[P_{X|Z}P_{Y|Z}\right] \| P_X P_Y\right)$ in Section C.1

- Proving $D_{\mathrm{KL}}\left(\mathbb{E}_{P_Z}\left[P_{X|Z}P_{Y|Z}\right] \| P_X P_Y\right) \leq H(Z)$ in Section C.2

- Proving $\mathrm{Cl} - \mathrm{InfoNCE}$ maximizes at $H(Z)$ when $H(Z|X) = H(Z|Y) = 0$ in Section C.3

## C.1    PART I - PROVING $\mathrm{Cl} - \mathrm{InfoNCE} \leq D_{\mathrm{KL}}\left(\mathbb{E}_{P_Z}\left[P_{X|Z}P_{Y|Z}\right] \| P_X P_Y\right)$

The proof requires the following lemma.

**Lemma C.3** (Theorem 1 by Song & Ermon (2020))**.** *Let $\mathcal{X}$ and $\mathcal{Y}$ be the sample spaces for $X$ and $Y$, $f$ be any function: $(\mathcal{X} \times \mathcal{Y}) \to \mathbb{R}$, and $\mathcal{P}$ and $\mathcal{Q}$ be the probability measures on $\mathcal{X} \times \mathcal{Y}$. Then,*

$$\sup_f \mathbb{E}_{(x,y_1)\sim\mathcal{P},(x,y_{2:n})\sim\mathcal{Q}^{\otimes(n-1)}}\left[\log \frac{e^{f(x,y_1)}}{\frac{1}{n}\sum_{j=1}^n e^{f(x,y_j)}}\right] \leq D_{\mathrm{KL}}\left(\mathcal{P} \| \mathcal{Q}\right).$$

Now, we are ready to prove the following lemma:

**Lemma C.4** (Proof Part I)**.** $\mathrm{Cl} - \mathrm{InfoNCE} := \sup_f \mathbb{E}_{(x_i,y_i)\sim\mathbb{E}_{z\sim P_Z}\left[P_{X|z}P_{Y|z}\right]^{\otimes n}}\left[\frac{1}{n}\sum_{i=1}^n \log \frac{e^{f(x_i,y_i)}}{\frac{1}{n}\sum_{j=1}^n e^{f(x_i,y_j)}}\right] \leq$
$D_{\mathrm{KL}}\left(\mathbb{E}_{P_Z}\left[P_{X|Z}P_{Y|Z}\right] \| P_X P_Y\right).$

*Proof.* By defining $\mathcal{P} = \mathbb{E}_{P_Z}\left[P_{X|Z}P_{Y|Z}\right]$ and $\mathcal{Q} = P_X P_Y$, we have

$$\mathbb{E}_{(x,y_1)\sim\mathcal{P},(x,y_{2:n})\sim\mathcal{Q}^{\otimes(n-1)}}\left[\log \frac{e^{f(x,y_1)}}{\frac{1}{n}\sum_{j=1}^n e^{f(x,y_j)}}\right] = \mathbb{E}_{(x_i,y_i)\sim\mathbb{E}_{z\sim P_Z}\left[P_{X|z}P_{Y|z}\right]^{\otimes n}}\left[\frac{1}{n}\sum_{i=1}^n \log \frac{e^{f(x_i,y_i)}}{\frac{1}{n}\sum_{j=1}^n e^{f(x_i,y_j)}}\right].$$

Plug in this result into Lemma C.3 and we conclude the proof. $\qquad\square$

## C.2    PART II - PROVING $D_{\mathrm{KL}}\left(\mathbb{E}_{P_Z}\left[P_{X|Z}P_{Y|Z}\right] \| P_X P_Y\right) \leq H(Z)$

The proof requires the following lemma:

**Lemma C.5.** $D_{\mathrm{KL}}\left(\mathbb{E}_{P_Z}\left[P_{X|Z}P_{Y|Z}\right] \| P_X P_Y\right) \leq \min\left\{\mathrm{MI}(Z;X), \mathrm{MI}(Z;Y)\right\}.$

*Proof.*

$$\mathrm{MI}(Z; X) - D_{\mathrm{KL}}\left(\mathbb{E}_{P_Z}\left[P_{X|Z}P_{Y|Z}\right] \| P_X P_Y\right)$$

$$= \int_z p(z) \int_x p(x|z) \log \frac{p(x|z)}{p(x)} \mathrm{d}x\mathrm{d}z - \int_z p(z) \int_x p(x|z) \int_y p(y|z) \log \frac{\int_{z'} p(z')p(x|z')p(y|z')\mathrm{d}z'}{p(x)p(y)} \mathrm{d}x\mathrm{d}y\mathrm{d}z$$

$$= \int_z p(z) \int_x p(x|z) \log \frac{p(x|z)}{p(x)} \mathrm{d}x\mathrm{d}z - \int_z p(z) \int_x p(x|z) \int_y p(y|z) \log \frac{\int_{z'} p(z'|y)p(x|z')\mathrm{d}z'}{p(x)} \mathrm{d}x\mathrm{d}y\mathrm{d}z$$

$$= \int_z p(z) \int_x p(x|z) \int_y p(y|z) \log \frac{p(x|z)}{\int_{z'} p(z'|y)p(x|z')\mathrm{d}z'} \mathrm{d}x\mathrm{d}y\mathrm{d}z$$

$$= -\int_z p(z) \int_x p(x|z) \int_y p(y|z) \log \frac{\int_{z'} p(z'|y)p(x|z')\mathrm{d}z'}{p(x|z)} \mathrm{d}x\mathrm{d}y\mathrm{d}z$$

$$\geq -\int_z p(z) \int_x p(x|z) \int_y p(y|z) \left(\frac{\int_{z'} p(z'|y)p(x|z')\mathrm{d}z'}{p(x|z)} - 1\right) \mathrm{d}x\mathrm{d}y\mathrm{d}z \ \left(\because \log t \leq t - 1\right)$$

$$= 0.$$

Hence, $\mathrm{MI}(Z; X) \geq D_{\mathrm{KL}}\left(\mathbb{E}_{P_Z}\left[P_{X|Z}P_{Y|Z}\right] \| P_X P_Y\right)$. Likewise, $\mathrm{MI}(Z; Y) \geq D_{\mathrm{KL}}\left(\mathbb{E}_{P_Z}\left[P_{X|Z}P_{Y|Z}\right] \| P_X P_Y\right)$. We complete the proof by combining the two results. $\square$

Now, we are ready to prove the following lemma:

**Lemma C.6** (Proof Part II). $D_{\mathrm{KL}}\left(\mathbb{E}_{P_Z}\left[P_{X|Z}P_{Y|Z}\right] \| P_X P_Y\right) \leq H(Z)$.

*Proof.* Combining Lemma C.5 and the fact that $\min\left\{\mathrm{MI}(Z; X), \mathrm{MI}(Z; Y)\right\} \leq H(Z)$, we complete the proof. Note that we consider $Z$ as the clustering assignment, which is discrete but not continuous. And the inequality holds for the discrete $Z$, but may not hold for the continuous $Z$. $\square$

## C.3 PART III - PROVING Cl − InfoNCE MAXIMIZES AT $H(Z)$ WHEN $H(Z|X) = H(Z|Y) = 0$

We directly provide the following lemma:

**Lemma C.7** (Proof Part III). $\mathrm{Cl} − \mathrm{InfoNCE}$ max. at $H(Z)$ when $H(Z|X) = H(Z|Y) = 0$.

*Proof.* When $H(Z|Y) = 0$, $p(Z|Y = y)$ is Dirac. The objective

$$D_{\mathrm{KL}}\left(\mathbb{E}_{P_Z}\left[P_{X|Z}P_{Y|Z}\right] \| P_X P_Y\right)$$

$$= \int_z p(z) \int_x p(x|z) \int_y p(y|z) \log \frac{\int_{z'} p(z')p(x|z')p(y|z')\mathrm{d}z'}{p(x)p(y)} \mathrm{d}x\mathrm{d}y\mathrm{d}z$$

$$= \int_z p(z) \int_x p(x|z) \int_y p(y|z) \log \frac{\int_{z'} p(z'|y)p(x|z')\mathrm{d}z'}{p(x)} \mathrm{d}x\mathrm{d}y\mathrm{d}z$$

$$= \int_z p(z) \int_x p(x|z) \int_y p(y|z) \log \frac{\int_{z'} p(z')p(x|z')p(y|z')\mathrm{d}z'}{p(x)p(y)} \mathrm{d}x\mathrm{d}y\mathrm{d}z$$

$$= \int_z p(z) \int_x p(x|z) \int_y p(y|z) \log \frac{p(x|z)}{p(x)} \mathrm{d}x\mathrm{d}y\mathrm{d}z = \mathrm{MI}\left(Z; X\right).$$

The second-last equality comes with the fact that: when $p(Z|Y = y)$ is Dirac, $p(z'|y) = 1 \ \forall z' = z$ and $p(z'|y) = 0 \ \forall z' \neq z$. Combining with the fact that $\mathrm{MI}\left(Z; X\right) = H(Z)$ when $H(Z|X) = 0$, we know $D_{\mathrm{KL}}\left(\mathbb{E}_{P_Z}\left[P_{X|Z}P_{Y|Z}\right] \| P_X P_Y\right) = H(Z)$ when $H(Z|X) = H(Z|Y) = 0$.

Furthermore, by Lemma C.4 and Lemma C.6, we complete the proof. $\square$

## C.4 Bringing Everything Together

We bring Lemmas C.4, C.6, and C.7 together and complete the proof of Theorem C.2.

## Appendix D  Algorithms

In this section, we provide algorithms for our experiments. We consider two sets of the experiments. The first one is K-means clusters + Cl-InfoNCE (see Section 4.4 in the main text), where the clusters involved in Cl-InfoNCE are iteratively obtained via K-means clustering on top of data representations. The second one is auxiliary-information-determined clusters + Cl-InfoNCE (see Section 4.3 in the main text and Section B.2), where the clusters involved in Cl-InfoNCE are pre-determined accordingly to data attributes (see Section 4.3 in the main text) or data hierarchy information (see Section B.2).

**K-means clusters + Cl-InfoNCE**  We present here the algorithm for K-means clusters + Cl-InfoNCE. At each iteration in our algorithm, we perform K-means Clustering algorithm on top of data representations for obtaining cluster assignments. The cluster assignment will then be used in our Cl-InfoNCE objective.

---
**Algorithm 1:** K-means Clusters + Cl-InfoNCE

---
**Result:** Pretrained Encoder $f_\theta(\cdot)$
$f_\theta(\cdot) \leftarrow$ Base Encoder Network;
Aug $(\cdot) \leftarrow$ Obtaining Two Variants of Augmented Data via Augmentation Functions;
Embedding $\leftarrow$ Gathering data representations by passing data through $f_\theta(\cdot)$;
Clusters $\leftarrow$ **K-means-clustering**(Embedding);
**for** *epoch in 1,2,...,N* **do**
  **for** *batch in 1,2,...,M* **do**
    data1, data2 $\leftarrow$ Aug(data_batch);
    feature1, feature2 $\leftarrow f_\theta$(data1), $f_\theta$(data2);
    $L_{\text{Cl-infoNCE}} \leftarrow$ Cl-InfoNCE(feature1, feature2, Clusters);
    $f_\theta \leftarrow f_\theta - lr * \frac{\partial}{\partial\theta}L_{\text{Cl-infoNCE}}$;
  **end**
  Embedding $\leftarrow$ gather embeddings for all data through $f_\theta(\cdot)$;
  Clusters $\leftarrow$ **K-means-clustering**(Embedding);
**end**

---

**Auxiliary information determined clusters + Cl-InfoNCE**  We present the algorithm to combine auxiliary-information-determined clusters with Cl-InfoNCE. We select data attributes or data hierarchy information as the auxiliary information, and we present their clustering determining steps in Section 3.1 in the main text for discrete attributes and Section B.1 for data hierarchy information.

---
**Algorithm 2:** Pre-Determined Clusters + Cl-InfoNCE

---
**Result:** Pretrained Encoder $f_\theta(\cdot)$
$f_\theta(\cdot) \leftarrow$ Base Encoder Network;
Aug $(\cdot) \leftarrow$ Obtaining Two Variants of Augmented Data via Augmentation Functions;
Clusters $\leftarrow$ Pre-determining Data Clusters from **Auxiliary Information**;
**for** *epoch in 1,2,...,N* **do**
  **for** *batch in 1,2,...,M* **do**
    data1, data2 $\leftarrow$ Aug(data_batch);
    feature1, feature2 $\leftarrow f_\theta$(data1), $f_\theta$(data2);
    $L_{\text{Cl-infoNCE}} \leftarrow$ Cl-InfoNCE(feature1, feature2, Clusters);
    $f_\theta \leftarrow f_\theta - lr * \frac{\partial}{\partial\theta}L_{\text{Cl-infoNCE}}$;
  **end**
**end**

---

## Appendix E    Experimental details

The following content describes our experiments settings in details. For reference, our code is available at `https://github.com/Crazy-Jack/Cl-InfoNCE/README.md`.

### E.1    UT-Zappos50K

The following section describes the experiments we performed on UT-Zappos50K dataset in Section 4 in the main text.

**Accessiblity**    The dataset is attributed to (Yu & Grauman, 2014) and available at the link: `http://vision.cs.utexas.edu/projects/finegrained/utzap50k`. The dataset is for non-commercial use only.

**Data Processing**    The dataset contains images of shoe from Zappos.com. We rescale the images to $32 \times 32$. The official dataset has 4 large categories following 21 sub-categories. We utilize the 21 subcategories for all our classification tasks. The dataset comes with 7 attributes as auxiliary information. We binarize the 7 discrete attributes into 126 binary attributes. We rank the binarized attributes based on their entropy and use the top-$k$ binary attributes to form clusters. Note that different $k$ result in different data clusters (see Figure 4 (a) in the main text).

*Training and Test Split*: We randomly split train-validation images by $7 : 3$ ratio, resulting in $35,017$ train data and $15,008$ validation dataset.

**Network Design**    We use ResNet-50 architecture to serve as a backbone for encoder. To compensate the 32x32 image size, we change the first 7x7 2D convolution to 3x3 2D convolution and remove the first max pooling layer in the normal ResNet-50 (See code for detail). This allows finer grain of information processing. After using the modified ResNet-50 as encoder, we include a 2048-2048-128 Multi-Layer Perceptron (MLP) as the projection head $\Big($i.e., $g(\cdot)$ in $f(\cdot, \cdot)$ equation (1) in the main text$\Big)$ for Cl-InfoNCE. During evaluation, we discard the projection head and train a linear layer on top of the encoder's output. For both K-means clusters + Cl-InfoNCE and auxiliary-information-determined clusters + Cl-InfoNCE, we adopt the same network architecture, including the same encoder, the same MLP projection head and the same linear evaluation protocol. In the K-means + Cl-InfoNCE settings, the number of the K-means clusters is $1,000$. Kmeans clustering is performed every epoch during training. We find performing Kmeans for every epoch benefits the performance. For fair comparsion, we use the same network architecture and cluster number for PCL.

**Optimization**    We choose SGD with momentum of $0.95$ for optimizer with a weight decay of $0.0001$ to prevent network over-fitting. To allow stable training, we employ a linear warm-up and cosine decay scheduler for learning rate. For experiments shown in Figure 4 (a) in the main text, the learning rate is set to be $0.17$ and the temperature is chosen to be $0.07$ in Cl-InfoNCE. And for experiments shown in Figure 5 in the main text, learning rate is set to be $0.1$ and the temperature is chosen to be $0.1$ in Cl-InfoNCE.

**Computational Resource**    We conduct experiments on machines with 4 NVIDIA Tesla P100. It takes about 16 hours to run 1000 epochs of training with batch size 128 for both auxiliary information aided and unsupervised Cl-InfoNCE.

### E.2    Wider Attributes

The following section describes the experiments we performed on Wider Attributes dataset in Section 4 in the main text.

**Accessiblity**    The dataset is credited to (Li et al., 2016) and can be downloaded from the link: `http://mmlab.ie.cuhk.edu.hk/projects/WIDERAttribute.html`. The dataset is for public and non-commercial usage.

**Data Processing**   The dataset contains $13,789$ images with multiple semantic bounding boxes attached to each image. Each bounding is annotated with $14$ binary attributes, and different bounding boxes in an image may have different attributes. Here, we perform the OR operation among the attributes in the bounding boxes in an image. Hence, each image is linked to $14$ binary attributes. We rank the $14$ attributes by their entropy and use the top-$k$ of them when performing experiments in Figure 4 (b) in the main text. We consider a classification task consisting of 30 scene categories.

*Training and Test Split*: The dataset comes with its training, validation, and test split. Due to a small number of data, we combine the original training and validation set as our training set and use the original test set as our validation set. The resulting training set contains $6,871$ images and the validation set contains $6,918$ images.

**Computational Resource**   To speed up computation, on Wider Attribute dataset we use a batch size of $40$, resulting in 16-hour computation in a single NVIDIA Tesla P100 GPU for $1,000$ epochs training.

**Network Design and Optimization**   We use ResNet-50 architecture as an encoder for Wider Attributed dataset. We choose 2048-2048-128 MLP as the projection head $\Big($ i.e., $g(\cdot)$ in $f(\cdot, \cdot)$ equation (1) in the main text$\Big)$ for Cl-InfoNCE. The MLP projection head is discarded during the linear evaluation protocol. Particularly, during the linear evaluation protocol, the encoder is frozen and a linear layer on top of the encoder is fine-tuned with downstream labels. For Kmeans + Cl-InfoNCE and Auxiliary information + Cl-InfoNCE, we consider the same architectures for the encoder, the MLP head and the linear evaluation classifier. For K-means + Cl-InfoNCE, we consider $1,000$ K-means clusters. For fair comparsion, the same network architecture and cluster number is used for experiments with PCL.

For Optimization, we use SGD with momentum of $0.95$. Additionally, $0.0001$ weight decay is adopted in the network to prevent over-fitting. We use a learning rate of $0.1$ and temperature of $0.1$ in Cl-InfoNCE for all experiments. A linear warm-up following a cosine decay is used for the learning rate scheduling, providing a more stable learning process.

E.3   CUB-200-2011

The following section describes the experiments we performed on CUB-200-2011 dataset in Section 4 in the main text.

**Accessiblity**   CUB-200-2011 is created by Wah et al. (2011) and is a fine-grained dataset for bird species. It can be downloaded from the link: http://www.vision.caltech.edu/visipedia/CUB-200-2011.html. The usage is restricted to non-commercial research and educational purposes.

**Data Processing**   The original dataset contains 200 birds categories over $11,788$ images with $312$ binary attributes attached to each image. We utilize those attributes and rank them based on their entropy, excluding the last $112$ of them (resulting in 200 attributes), because including these $112$ attributes will not change the number of the clusters than not including them. In Figure 4 (c), we use the top-$k$ of those attributes to constrcut clusters with which we perform in Cl-InfoNCE. The image is rescaled to $224 \times 224$.

*Train Test Split*: We follow the original train-validation split, resulting in $5,994$ train images and $5,794$ validation images.

**Computational Resource**   It takes about 8 hours to train for 1000 epochs with 128 batch size on 4 NVIDIA Tesla P100 GPUs.

**Network Design and Optimization**   We choose ResNet-50 for CUB-200-2011 as the encoder. After extracting features from the encoder, a 2048-2048-128 MLP projection head $\Big($ i.e., $g(\cdot)$ in $f(\cdot, \cdot)$ equation (1) in the main text$\Big)$ is used for Cl-InfoNCE. During the linear evaluation protocal, the MLP

projection head is removed and the features extracted from the pre-trained encoder is fed into a linear classifier layer. The linear classifier layer is fine-tuned with the downstream labels. The network architectures remain the same for both K-means clusters + Cl-InfoNCE and auxiliary-information-determined clusters + Cl-InfoNCE settings. In the K-means clusters + Cl-InfoNCE settings, we consider $1,000$ K-means clusters. For fair comparsion, the same network architecture and cluster number is used for experiments with PCL.

SGD with momentum of $0.95$ is used during the optimization. We select a linear warm-up following a cosine decay learning rate scheduler. The peak learning rate is chosen to be $0.1$ and the temperature is set to be $0.1$ for both K-means + Cl-InfoNCE and Auxiliary information + Cl-InfoNCE settings.

### E.4    IMAGENET-100

The following section describes the experiments we performed on ImageNet-100 dataset in Section 4 in the main text.

**Accessibility**    This dataset is a subset of ImageNet-1K dataset, which comes from the ImageNet Large Scale Visual Recognition Challenge (ILSVRC) 2012-2017 (Russakovsky et al., 2015). ILSVRC is for non-commercial research and educational purposes and we refer to the ImageNet official site for more information: https://www.image-net.org/download.php.

**Data Processing**    In the Section 4 in the main text and Section B, we select 100 classes from ImageNet-1K to conduct experiments (the selected categories can be found in https://github.com/Crazy-Jack/Cl-InfoNCE/data_processing/imagenet100/selected_100_classes.txt). We also conduct a slight pre-processing (via pruning a small number of edges in the WordNet graph) on the WordNet hierarchy structure to ensure it admits a tree structure. Specifically, each of the selected categories and their ancestors only have one path to the root. We refer the pruning procedure in https://github.com/Crazy-Jack/Cl-InfoNCE/data_processing/imagenet100/hierarchy_processing/imagenet_hierarchy.py (line 222 to 251).

We cluster data according to their common ancestor in the pruned tree structure and determine the level $l$ of each cluster by the step needed to traverse from root to that node in the pruned tree. Therefore, the larger the $l$, the closer the common ancestor is to the real class labels, hence more accurate clusters will be formed. Particularly, the real class labels is at level 14.

*Training and Test Split*: Please refer to the following file for the training and validation split.

- training: https://github.com/Crazy-Jack/Cl-InfoNCE/data_processing/imagenet100/hier/meta_data_train.csv
- validation: https://github.com/Crazy-Jack/Cl-InfoNCE/data_processing/imagenet100/hier/meta_data_val.csv

The training split contains $128,783$ images and the test split contains $5,000$ images. The images are rescaled to size $224 \times 224$.

**Computational Resource**    It takes 48-hour training for 200 epochs with batch size 128 using 4 NVIDIA Tesla P100 machines. All the experiments on ImageNet-100 is trained with the same batch size and number of epochs.

**Network Design and Optimization Hyper-parameters**    We use conventional ResNet-50 as the backbone for the encoder. 2048-2048-128 MLP layer and $l2$ normalization layer is used after the encoder during training and discarded in the linear evaluation protocal. We maintain the same architecture for Kmeans + Cl-InfoNCE and auxiliary information aided Cl-InfoNCE. For Kmeans + Cl-InfoNCE, we choose 2500 as the cluster number. For fair comparsion, the same network architecture and cluster number is used for experiments with PCL. The Optimizer is SGD with $0.95$ momentum. For K-means + Cl-InfoNCE used in Figure 5 in the main text, we use the learning rate of $0.03$ and the temperature of $0.2$. We use the learning rate of $0.1$ and temperature of $0.1$ for auxiliary information + Cl-InfoNCE in Figure 6. A linear warm-up and cosine decay is used for the learning

rate scheduling. To stablize the training and reduce overfitting, we adopt 0.0001 weight decay for the encoder network.

## APPENDIX F    COMPARISONS WITH SWAPPING CLUSTERING ASSIGNMENTS BETWEEN VIEWS

In this section, we provide additional comparisons between Kmeans + Cl-InfoNCE and Swapping Clustering Assignments between Views (SwAV) (Caron et al., 2020). The experiment is performed on ImageNet-100 dataset. SwAV is a recent art for clustering-based self-supervised approach. In particular, SwAV adopts Sinkhorn algorithm (Cuturi, 2013) to determine the data clustering assignments for a batch of data samples, and SwAV also ensures augmented views of samples will have the same clustering assignments. We present the results in Table 5, where we see SwAV has similar performance with the Prototypical Contrastive Learning method (Li et al., 2020) and has worse performance than our method (i.e., K-means +Cl-InfoNCE).

| Method | Top-1 Accuracy (%) |
|---|---|
| *Non-clustering-based Self-supervised Approaches* | |
| SimCLR (Chen et al., 2020) | 58.2±1.7 |
| MoCo (He et al., 2020) | 59.4±1.6 |
| *Clustering-based Self-supervised Approaches (# of clusters = 2.5K)* | |
| SwAV (Caron et al., 2020) | 68.5±1.0 |
| PCL (Li et al., 2020) | 68.9±0.7 |
| K-means + Cl-InfoNCE (ours) | **77.9±0.7** |

Table 5: Additional Comparsion with SwAV (Caron et al., 2020) showing its similar performance as PCL on ImageNet-100 dataset.

## APPENDIX G    PRELIMINARY RESULTS ON IMAGENET-1K WITH CL-INFONCE

We have performed experiments on ImageNet-100 dataset, which is a subset of the ImageNet-1K dataset (Russakovsky et al., 2015). We use the batch size of $1,024$ for all the methods and consider 100 training epochs. We present the comparisons among Supervised Contrastive Learning (Khosla et al., 2020), our method (i.e., WordNet-hierarchy-information-determined clusters + Cl-InfoNCE), and SimCLR (Chen et al., 2020). We select the level-12 nodes in the WordNet tree hierarchy structures as our hierarchy-determined clusters for Cl-InfoNCE. We report the results in Table 6. We find that our method (i.e., hierarchy-determined clusters + Cl-InfoNCE) performs in between the supervised representations and conventional self-supervised representations.

| Method | Top-1 Accuracy (%) |
|---|---|
| *Supervised Representation Learning (Z = downstream labels T)* | |
| SupCon (Khosla et al., 2020) | 76.1±1.7 |
| *Weakly Supervised Representation Learning (Z = level 12 WordNet hierarchy labels)* | |
| Hierarchy-Clusters + Cl-InfoNCE (ours) | 67.9±1.5 |
| *Self-supervised Representation Learning (Z = instance ID)* | |
| SimCLR (Chen et al., 2020) | 62.9±1.2 |

Table 6: Preliminary results for WordNet-hierarchy-determined clusters + Cl-InfoNCE on ImageNet-1K.

## APPENDIX H  SYNTHETICALLY CONSTRUCTED CLUSTERS IN SECTION 4.2 IN THE MAIN TEXT

In Section 4.2 in the main text, on the UT-Zappos50K dataset, we synthesize clusters $Z$ for various $I(Z;T)$ and $H(Z|T)$ with $T$ being the downstream labels. There are 86 configurations of $Z$ in total. Note that the configuration process has no access to data's auxiliary information and among the 86 configurations we consider the special cases for the supervised $\big(Z = T\big)$ and the unsupervised setting $\big(Z = \text{instance ID}\big)$. In specific, when $Z = T$, $I(Z;T)$ reaches its maximum at $H(T)$ and $H(Z|T)$ reaches its minimum at 0; when $Z = \text{instance ID}$, both $I(Z;T)$ $\big(\text{to be } H(T)\big)$ and $H(Z|T)$ $\big(\text{to be } H(\text{instance ID})\big)$ reaches their maximum. The code for generating these 86 configurations can be found in lines 177-299 in https://github.com/Crazy-Jack/Cl-InfoNCE/data_processing/UT-zappos50K/synthetic/generate.py.

