# OpenReview forum: "Learning Weakly-supervised Contrastive Representations"
_ICLR.cc/2022/Conference — ICLR 2022 Poster_

### Official Review · Reviewer_1ziy · 2021-10-31

**Correctness:** 3
**Technical Novelty And Significance:** 2
**Empirical Novelty And Significance:** 2
**Recommendation:** 5
**Confidence:** 4

**Main Review:**

Strengths:
- The experiments are well-organized and are clearly presented. The sub-sections make it clear which experiments refer to which claims by the authors.
- The experiments are on a wide-set of image datasets, and on both weakly-supervised and unsupervised versions of the proposed method, which helps show the generalization ability of the method. (I'd like to see more investigation of the labels though, see below.)

Weaknesses:
- My main concern is that the paper does not sufficiently address other relevant works using weakly supervised contrastive loss, and so the contribution of this work is not clear. More specifically, there are other works that also use auxiliary information/weak-supervision with contrastive loss [A] (the proposed Eq (1) in this paper seems similar to the weakly supervised contrastive loss in Eq (4) of [A], both using additional weak supervision on contrastive loss (in contrast to fully supervised contrastive loss as in (Khosla et al., 2020))). Another set of works are other clustering-based contrastive loss methods [B] (On ImageNet100, looks like the contrastive loss proposed in [B] performs better than this method. Table 2 in [B] = 79.77, Table 5 here = 77.9). Comparing against these as existing baselines would be best, or at least there should be a discussion to contrast with such existing works.
- The performance of this method seems limited relative to existing works - Table 2 shows that CMC is similar in performance to the proposed method. The authors explain that "Cl-InfoNCE performs better with less informative auxiliary information": it would make the paper stronger if the authors specify more concretely when their method should be used over CMC. Also [B] seems to perform better than this method on ImageNet100.
- What if the auxiliary information is incorrect or noisy? Would this method perform worse than SimCLR, or will the method learn to ignore the incorrect label information?
- Minor: The paper would be stronger if it discussed other works using auxiliary information or weak supervision on data, since using auxiliary information is a major part of this paper. Adding a paragraph discussing this would be great (ex: works like [C]) so that the contribution of this paper is more clear.
- Minor: Minor issues with formatting - for example, the plot in Figure 5 looks squished. Also, the writing could be more clear in places (ex: "For instance, if having continuous attributes as auxiliary information, binning or quantization cannot be avoided when constructing the clusters." my understanding is that the authors are saying, the proposed method cannot work with continuous attribute labels?). The Appendix also appears to be submitted with the main PDF (instead of being in supp materials).

[A] Sun, Jennifer J., et al. "Task programming: Learning data efficient behavior representations." CVPR 2021.

[B] Zheng, Mingkai, et al. "Weakly Supervised Contrastive Learning." ICCV 2021.

[C] Tan, Reuben, et al. "Learning similarity conditions without explicit supervision." CVPR 2019.

**Summary Of The Paper:**

The paper proposes a weakly-supervised contrastive loss based on using existing auxiliary information from datasets (such as shoe attributes for classifying shoe categories (UT-zappos50K)). The auxiliary information is used in a contrastive loss formulation similar to the unsupervised contrastive loss of (Chen et al., 20) and the supervised contrastive loss formulation of (Khosla et al., 2020), but instead using clustering information from auxiliary information (Eq (1) of the paper). Experiments are across 4 image datasets (UT-zappos50K, Wider Attribute, CUB-200-2011, ImageNet-100) and shows that representations learned from the proposed weakly-supervised loss performs between unsupervised & fully supervised contrastive loss. There are additional experiments comparing to a few other clustering-based contrastive loss approaches.

**Summary Of The Review:**

Overall, the experimental section of the paper was clear and included a good variety of image classification datasets. However, given the lack of discussion/comparison to relevant existing works and limited performance of the proposed method, the contribution of the proposed method is not clear to me at this point.

---

> ### Author Response · Authors · 2021-11-22
> **Response to Reviewer 1ziy**
>
>
> We thank the reviewer for the comments. Please see the following response. Please let us know if any of the concerns are not addressed by our response.
>
> **(Comparisons with Other Baselines)** In the related work section (Section 2 in our submission), we did provide discussions with other related work that utilize the auxiliary information of data. The discussed related work ("learning to predict the auxiliary information using cross-entropy loss" and "multi-view contrastive learning") are all been compared in our experimental section.
>
> The paper [1] suggested by the reviewer has a very different setting (focuses on behavior modeling) from ours, and we humbly argue that this paper is out of the scope of our paper. How this paper utilizes the auxiliary information is to predict the auxiliary information from the representations. This loss has been considered as a baseline in our experimental section: in precise, the Attributes-Determined Clusters + Cross-Entropy Loss method in Table 2. We also like to make a remark that the paper [2] appears publicly available on the last day of ICLR submission. For the performance difference on ImageNet-100, we acknowledge that we have different versions of ImageNet-100, batch size, and architectures. We are happy to provide comparisons with [2] both conceptually and experimentally (please refer to the general response in this rebuttal). Last, the paper [3] suggested by the reviewer falls in the category of learning to predict the auxiliary information. We will add this paper [3] into our related work discussion.
>
> **(Remark on Contribution)** To be clear on our contribution, we introduce a new framework to consider structural information within contrastive learning, and this structural information can be derived from data attributes, data hierarchy structure (details are in Supplementary), or even the data themselves (unsupervised). We also present information-theoretical analysis to explain how different structural information can affect the learned representations.
>
> **(Cl-InfoNCE v.s. CMC)** We thank the reviewer to bring up this great question: when to use Cl-InfoNCE and when to use CMC. We will include the following response in our revised manuscript. To answer this question, we recall that CMC learns embeddings from both the input data and their auxiliary information to perform contrastive learning. When the auxiliary information is low-dimensional or contains too much noise, we can expect learning an embedding for it will suffer from overfitting and further affect the downstream performance of the learned representations. To see this, in Table2 in our submission, we see that CMC performs much worse on the Wider dataset compared to the performance on CUB datasets. For these two datasets, Wider has only 14 attributes whereas CUB has around 200 attributes. To conclude, if the auxiliary information is low-dimensional or we are not sure how noisy it is, we should go for the Cl-InfoNCE method.
>
> **(Remark on Noisy Auxiliary Information)** This is another great question brought up by the reviewer. In fact, the auxiliary information is always noisy. For instance, in all of our experiments, $H(Z|T)\neq 0$, where $H(Z|T)$ measures how much redundant information contains in the auxiliary information but not in the downstream labels. In addition, $I(Z; T)$ is not always at its maximum which is $I(T; T)$ if our auxiliary information $Z$ is as perfect as $T$. These measurements indicate that the auxiliary information we used is always noisy with respect to the downstream labels. However, our method still manages to utilize the noisy auxiliary information and learn better representations.
>
> **(Remark on Continuous Attribute Labels)** Our method performs contrastive learning based on data clusters. The data clusters are discrete in nature, and hence a direct way to work with continuous attribute labels is to perform quantization and binning.
>
> **(Appendix Should be in the Main PDF)** We checked our original submission and the appendix is attached at the end of our main pdf. The Appendix in supplementary material is the same one, and we will remove it.
>
>
> [1] Sun, Jennifer J., et al. "Task programming: Learning data efficient behavior representations." CVPR 2021.
>
> [2] Zheng, Mingkai, et al. "Weakly Supervised Contrastive Learning." ICCV 2021.
>
> [3] Tan, Reuben, et al. "Learning similarity conditions without explicit supervision." CVPR 2019.

---

> > ### Comment · Reviewer_1ziy · 2021-11-29
> > **Thanks for your update.**
> >
> > I appreciate the authors for updating their paper and the responses. I understand the comparison with [2] and vs. CMC, and the experiments are promising. Therefore, the contributions are more clear to me, however I would like further clarifications on the main claims vs. existing work:
> >
> > > we introduce a new framework to consider structural information within contrastive learning, and this structural information can be derived from data attributes, data hierarchy structure (details are in Supplementary), or even the data themselves (unsupervised).
> >
> > Since this is the contribution claimed by the authors, it is very similar to [1] in terms of the claims and losses (specifically the first claim where structural information is derived from data attributes for contrastive loss). The authors mention that [1] studies the "Attributes-Determined Clusters + Cross-Entropy Loss method in Table 2", corresponding to Eq (3) from [1] ("attribute decoding"); however, the authors did not address that additionally, "Attributes-Determined Clusters + Cl-InfoNCE" corresponds to Eq (4) from [1] ("contrastive loss"). Adding a discussion on how the proposed loss is different from the contrastive loss in Eq (4) of [1] or otherwise comparing the two frameworks would be helpful.
> >
> > While the experimental settings are different, the losses under study are similar, and reviewer a8kw further suggests that "These methods are straightforward and common tricks in representation learning literature.". It would be good to have further comments from the authors on novelty of the method (since that is part of the claims of the paper), to backup (or modify) the claim that the framework is new, considering such structured contrastive losses have been studied previously.
> >
> > [1] Sun, Jennifer J., et al. "Task programming: Learning data efficient behavior representations." CVPR 2021.
> >
> > [2] Zheng, Mingkai, et al. "Weakly Supervised Contrastive Learning." ICCV 2021.

---

> > > ### Author Response · Authors · 2021-11-30
> > > **Reviewer's Response**
> > >
> > > We thank the reviewer for bringing this out. The difference between eq(4) from [1] and ours is how the positives/negatives are selected. Eq(4) from [1] selects the positives for instances from the same task and negatives from different tasks. On the other hand, we select the positives for instances from the same cluster and negatives from different clusters. Tasks and clusters are different in notions but do share similarities. We will include this remark in the final manuscript; as of now, we are not able to update the manuscript.
> > >
> > > Our paper has three main components: our formulation of clustering-based contrastive learning, the attempt to incorporate side information into contrastive learning, and the information-theoretical analysis. The major concern from the reviewer is that our formulation of clustering-based contrastive learning is not completely novel, and we humbly accept this criticism. However, aside from our formulation, we hope the other two components in our paper can be recognized: the attempt of incorporating side information into contrastive learning and the information-theoretical analysis for clustering-based contrastive learning. We believe these are new and can benefit the community for contrastive learning.
> > >
> > > [1] Sun, Jennifer J., et al. "Task programming: Learning data efficient behavior representations." CVPR 2021

---

> > > > ### Comment · Reviewer_1ziy · 2021-11-30
> > > > **Thanks for your response.**
> > > >
> > > > My understanding is that in [1], each task (behavior attribute) is analogous to an attribute here. While [1] essentially forms different sets of clusters for every task, here the authors use the method in Section 3.1 to select attributes for clustering; as such, Eq (4) in [1] is very similar to Eq (1) in this work since the form the contrastive loss is the same.
> > > >
> > > > To be clear, my main concern with this paper is not the novelty of the method, but rather the accuracy of the author's claims (in terms of the novelty claimed by the authors of incorporating auxiliary into contrastive loss as I mentioned above, also shared by other reviewers). The "clustering-based contrastive learning", and "the attempt to incorporate side information into contrastive learning" has been studied by previous works as we've discussed (for me, it is ok if these are studied by other works previously (since the paper has other theoretical/empirical results), but existing works in the space should be mentioned and discussed/compared in the paper by the authors to show where the contribution of this specific work is). Again, my concern is not the novelty itself, but the accuracy of claims - for future paper versions, I hope the authors will more accurately and clearly describe their contributions.
> > > >
> > > > [1] Sun, Jennifer J., et al. "Task programming: Learning data efficient behavior representations." CVPR 2021

---

> > > > > ### Author Response · Authors · 2021-11-30
> > > > > **Reviewer's Response**
> > > > >
> > > > > Thanks for the clarification! After the reviewer's explanation, we now understand why the paper [1] can be understood as bringing side information into contrastive learning. We sincerely appreciate the sharing of the reviewer, and we would be happy to revise the wordings of contributions accordingly:
> > > > >
> > > > > "We present building blocks to incorporate auxiliary information or side information into contrastive learning objectives. Different from prior work, our framework allows flexible forms of auxiliary information to be considered in contrastive learning, such as auxiliary attributes (studied in the main text) or hierarchy information (studied in Supplementary). The idea behind our building blocks is to form clusters for auxiliary/side information then consider a clustering-based contrastive objective. To have a better understanding of our method, we also present information-theoretical analysis to study the presented clustering-based contrastive objective."

---

### Official Review · Reviewer_a8kw · 2021-11-01

**Correctness:** 3
**Technical Novelty And Significance:** 2
**Empirical Novelty And Significance:** 2
**Recommendation:** 5
**Confidence:** 4

**Main Review:**

STRENGTHS:

++ Interesting idea on leveraging the structure information hidden from the data distribution for learning better representation.

++ Thorough theoretical analysis on the value of clustering information in the proposed objective function.

++ Well written paper.

WEAKNESSES:

-- Lack of novelty. The proposed Cl-InfoNCE is simply a cluster aggregated NCE loss. The way to construct the cluster is either through top-k attributes or k-means. These methods are straightforward and common tricks in representation learning literature.

-- Weakly-supervised methods refer to those using noise labels for learning certain tasks. In the context of this paper, it is essentially the using the auxiliary information as structure information to guide the learning process, and the labels utilized are the accurate ground truth. In my opinion, it is misleading to use the term “weakly supervised”, and the method itself is more of  like “structure aware representation learning”.

-- Some key details are not clarified. The attribute-determined clusters contain samples with the same values on a set of attributes, which inevitably produce some sparse clusters with a very small number of samples. How does that affect the model performance ? Is overfitting an issue in such a situation? How to handle the sample imbalance across different clusters ? Though not mentioned, such practical issues would greatly affect the model performance.

-- Experiments are not sufficient. i). There are other similar existing works with higher performance. For example, Zheng et al. Weakly Supervised Contrastive Learning, ICCV 2021. (79.7% vs 77.9 (this paper) on ImageNet-100). ii) How does the proposed method work compared with other structure-aware representation learning methods such as “Yaling Tao, et al. Clustering-friendly Representation Learning via Instance Discrimination and Feature Decorrelation, ICLR 2021”.


**Summary Of The Paper:**

This paper proposes a two-stage weakly supervised contrastive learning approach, where it first clusters the data according to the available auxiliary information (or data -driven clusters if no auxiliary information available), and then generates similar representations for the intra-cluster samples and dissimilar representations for inter-cluster samples via a clustering InfoNCE objective. Theoretically, the objective is proved to include the clustering information in the learned representation, and can serve as an interpolation between supervised and self-supervised representation learning. Experiments on four datasets with and without auxiliary information validate the effectiveness of the proposed method.


**Summary Of The Review:**

This paper proposes a structure-aware constructive loss with theoretical justification for representation learning. The theoretical analysis is very interesting and insightful. However, there are some issues in terms of novelty, clarity in the technical details, rationality on the terminology as well as the soundness in the experiments.

---

> ### Author Response · Authors · 2021-11-21
> **Response to Reviewer a8kw - Part 2**
>
> **(Comparisons with Other Baselines)** The major concern from the reviewer is the comparisons with other baselines. We like to make a remark that the paper [1] appears publicly available on the last day of ICLR submission. For the performance difference on ImageNet-100, we acknowledge that we have different versions of ImageNet-100, batch size, and architectures. We are happy to provide comparisons with [1] both conceptually and experimentally (please refer to the general response in this rebuttal). We also thank the reviewer for bringing another related paper [2], and we are happy to provide comparisons with it (the discussion will be put in the revised manuscript). [2] presents to learn representations that are clustering friendly (from a spectral clustering viewpoint). The main difference between [2] and ours is that [2] does not utilize the constructed clusters as weak labels during contrastive training, while ours regards the constructed clusters as weak supervision signals for the contrastive objective. To provide a fair comparison with [2], we stick to the training paradigm in [2] that replaces Resnet-50 with Resnet-18 for our method with the same batch size of 128. Note that [2] was focusing on clustering quality and didn't report the linear evaluation protocol. Hence we use the released code of [2] and experiments on CIFAR10. Both linear evaluation protocol and clustering accuracy metrics are used. Due to the time constraint, we train both methods for 1000 epochs for a fair comparison. The results are presented below.
>
>
> | Method | Linear Evaluation (%) | Clustering Accuracy (%) | Training Epochs|
> | -------- | -------- | -------- | -------- |
> | IDFD [2]   | 81.7 +- 0.9  | 66.9 +- 1.4 | 550 |
> | Cl-InfoNCE + Kmeans (ours) | **89.5 +- 0.6** | **78.3 +- 1.9** | 550|
> | IDFD [2]   | 82.1 +- 1.2  | 70.1 +- 1.7 | 750 |
> | Cl-InfoNCE + Kmeans (ours) | **90.2 +- 0.7** | **79.1 +- 2.1** | 750|
> | IDFD [2]   | 84.4 +- 1.1  | 77.7 +- 1.5 | 1000 |
> | Cl-InfoNCE + Kmeans (ours) | **90.7 +- 0.8** | **81.1 +- 1.2** | 1000|
>
>
> Note that [2] proposed 2 methods (IDFD and IDFO), we choose the compare with IDFD because 1) IDFO is very unstable, 2) IDFD/IDFO perform at-par for the best performance based on Figure2 in [2] and 3) [2] only officially released code for IDFD. For our method, we tested various cluster numbers for the Kmeans step and found K=1000 works for the best.
>
> From the results above we can see that Cl-InfoNCE + Kmeans facilitates faster convergence as well as higher performance (for both linear evaluation and clustering accuracy measurements).
>
> [1] Weakly Supervised Contrastive Learning. ICCV 2021.
>
> [2] Clustering-friendly Representation Learning via Instance Discrimination and Feature Decorrelation. ICLR 2021.

---

> ### Author Response · Authors · 2021-11-21
> **Response to Reviewer a8kw - Part 1**
>
> We thank the reviewer for the comments. Please see the following response. Please let us know if any of the concerns are not addressed by our response.
>
> **(Remarks on Novelty and Contribution)** We agree with the reviewer that selecting top-k attributes or using K-means clustering as one piece in our method is straightforward. To be clear on our contribution, we introduce a new framework to consider structural information within contrastive learning, and this structural information can be derived from data attributes, data hierarchy structure (details are in Supplementary), or even the data themselves (unsupervised). We also present information-theoretical analysis to explain how different structural information can affect the learned representations. Although the building blocks of our method exist previously, our work provides a new perspective of how effectively incorporate general structural information into contrastive representation learning.
>
>
> **(Confusion of using the Term Weakly-supervised)** We thank the reviewer for suggesting "structure-aware" for our method, and we agree this term is more precise than "weakly-supervised". Nonetheless, we argue that as long as the supervision we used is noisy or imperfect with respect to the downstream labels, it can be counted as "weakly-supervised". For instance, in all of our experiments, $H(Z|T)\neq 0$, which $H(Z|T)$ measures how much redundant information contains in the constructed labels but not in the downstream labels. The excess information indicates that the constructed labels we used are noisy with respect to the downstream labels.
>
>
> **(Sparse Clusters with Small Number of Samples)** We thank the reviewer to bring up this great question. After further investigation, the imbalance cluster issue sometimes appears (not often) when the attributes are selected randomly, and selecting the attributes with high entropy (what we proposed) effectively avoids this issue. In precise, we observe that selecting the attributes with high entropy makes the number of data per cluster quite evenly. Moreover, we do find that the imbalance cluster issue harms the performance, as the reviewer argued. To remedy this issue, other than re-clustering the data, we find that simply removing the sparse clusters in our training set can bring back the performance.

---

> > ### Comment · Reviewer_a8kw · 2021-11-29
> > **Discussion on Authors' Responses**
> >
> > Thanks for the author's responses! The new experiments show that the proposed method outperform other structure-aware baselines in [1] (marginally) and [2].
> >
> > However, my major concern on the technical novelty still remains. As the author claimed in their responses, the main contribution of this work is to introduce the structure information into the contrastive learning by deriving it from data attributes, data clustering and so on. In my opinion, such kinds of derivation are commonly-used tricks in the existing works and the entire framework still fall into the classic contrastive learning. The only difference is just to use cluster assignment to aggregate the NCE loss, which seems not significant enough to the research community.

---

> > > ### Author Response · Authors · 2021-11-29
> > > **Reviewer's Response**
> > >
> > > Thanks for the response! We are sorry to hear that the reviewer regards our work as an incremental aggregation between cluster assignment and the NCE loss. We humbly accept this critism. However, aside from the aggregation, we hope our attempt that brings side information of data within contrastive learning and the information-theoretical analysis of clustering-based contrastive learning could be recognized. We believe these are new and can benefit the community.

---

### Official Review · Reviewer_h3zd · 2021-11-02

**Correctness:** 4
**Technical Novelty And Significance:** 3
**Empirical Novelty And Significance:** 3
**Recommendation:** 6
**Confidence:** 4

**Main Review:**

Strengths：
- The idea is clear and simple, and the authors give theoretical definitions to support their assumptions and experiments.
- The authors prove that Cl-InfoNCE maximization learns to include the clustering information. And thus can use I(Z; T) and H(Z|T) to characterize the goodness of the learned representations. The authors also present experiments to validate it.
- The authors conduct experiments with different supervision levels and show that CL-InfoNCE can better bridge the gap with the supervised learned representations by using auxiliary information.
- The proposed approach can also be applied without auxiliary information with K-means and EM optimization.

Weaknesses:
- In Figure 4, the authors observe that the best performing clusters happen at the intersection between I(Z;T) and negative H(Z|T), could you present proves on this?
- Comparison with related work is required, such as [1]

[1] Weakly Supervised Contrastive Learning. ICCV 2021.

**Summary Of The Paper:**

This paper proposes a weakly-supervised contrastive representation by using the auxiliary clustering information. Data are clustered with auxiliary tags and the clustering InfoNCE loss is introduced to optimize the system. The authors prove that maximizing the Cl-InfoNCE learns to include the clustering information.

**Summary Of The Review:**

Overall, this paper proposes an easy but effective method for weakly-supervised contrastive learning with auxiliary information. The authors provide clear mathematical proof and good results with adequate experiments. However, comparisons with related work should be included to make to final results more convincing.

---

> ### Author Response · Authors · 2021-11-21
> **Response to Reviewer h3zd**
>
>
> We thank the reviewer for the comments. Please see the following response. Please let us know if any of the concerns are not addressed by our response.
>
> **(Proof of Optimality between the Intersection between $I(Z;T)$ and $-H(Z|T)$)** Unfortunately, we do not have proof for this observation. In the submission, we carefully stated this is an empirical observation, and we use this empirical observation to determine the hyper-parameter - number of clusters with the training and the validation set.
>
> **(Comparison with [1])** We do not have knowledge about [1] during our submission since [1] is publicly available on the last day of ICLR's submission. Nonetheless, we are happy to provide comparisons with [1] both conceptually and experimentally (see the general response of this rebuttal).
>
> [1] Weakly Supervised Contrastive Learning. ICCV 2021.

---

### Official Review · Reviewer_RhYi · 2021-11-02

**Correctness:** 3
**Technical Novelty And Significance:** 3
**Empirical Novelty And Significance:** 3
**Recommendation:** 8
**Confidence:** 3

**Main Review:**

Strengths:
- The paper is very interesting.
- The authors conduct a large number of experiments, and nearly all of them seem to support the authors' main claims.
- The results in Figure 4 are particularly impressive.

Weaknesses:
- Fairly minor:
  - Some of the presentation could be a bit clearer. For example, Eqn (1) could make clearer how the negatives are drawn.
  - In 4.3 the authors mention that their approach is more stable/robust to hyperparameters than the cross-ent baseline, and it would be nice to have a bit more detail about what exactly the authors observed.

Additional comment/question:
I'm wondering if the authors have an intuition about why exactly the K-Means+Cl-InfoNCE approach is helpful. Should we think of the initial pass through f and the subsequent clustering as clustering the data based on random features? If so, is it actually necessary to alternate K-means and Cl-InfoNCE, or can we just use the initial random clustering?

**Summary Of The Paper:**

The paper proposes an objective for contrastive training of representations based on noisy cluster information. In particular, the objective encourages samples from the same noisy cluster to have similar representations and representations from distinct clusters to have dissimilar representations. The authors show that representations learned with this approach often outperform representations learned with only self-supervision (and no noisy clusters), and also that the proposed approach to using noisy cluster information outperforms other baselines using noisy cluster information. The authors further show that they are able to alternate representation learning and clustering in order to learn better representations that self-supervised approaches even without initial noisy clusters.


**Summary Of The Review:**

This is an interesting paper with results that do an impressive job of supporting the paper's claims.

---

> ### Author Response · Authors · 2021-11-21
> **Response to Reviewer RhYi**
>
>
> We thank the reviewer for the comments. Please see the following response. Please let us know if any of the concerns are not addressed by our response.
>
> **(Negatives in Eq. (1))** In Eq. (1), $Z$ represents the weak labels of data, such as the weak labels determined by the data attributes or the K-means clustering approach. We first sample several different weak labels, and for each weak label, we sample some data points. The positives are drawn between data with the same weak labels, and the negatives are drawn between data with different weak labels. In short, our sampling process ensures the positively-paired data has the same weak label and the negatively-paired data has different weak labels.
>
> **(More Robust to Hyper-parameters than the Cross-entropy Method)** We have a similar observation with the paper [1], which *"adopting contrastive learning with labels"* is more robust to hyper-parameters than *"adopting cross-entropy loss with labels"*. In particular, we find the final performance of contrastive learning is less sensitive to the optimizer choice (e.g., Adam or SGD) and the learning rate (e.g., 1e-3 or 5e-4).
>
> **(Remarks on K-means Clustering)** This is a great question. We did try two different settings: 1) initializes the K-means clusters on random features from data and then fixes the K-means clusters afterward, and 2) iterates between K-means clusters construction and Cl-InfoNCE. We find the former does not perform as well as the latter, and we argue the reason is that iterating cluster construction can make the clusters better. Nonetheless, it does not mean the former does not work at all, where its result can be as good as SimCLR (a strong baseline for unsupervised contrastive representation learning). We argue that exploring implicit or explicit structural information within data for representation learning has a strong potential and is not fully explored yet.
>
> [1] Supervised Contrastive Learning. NIPS 2020.

---

### Author Response · Authors · 2021-11-21
**General Response**

We appreciate the reviewer's comments and suggestions. We notice that the paper [1] is brought up by multiple reviewers with the criticisms of 1) not discussing it and 2) it has a better performance on ImageNet-100. We would like to note that the paper [1] is a concurrent work with ours. The paper is only made publicly available on 10/05/2021, which is the same day as the paper submission deadline for ICLR'22. To be more precise, ICCV publicly released this paper on 10/05/2021, and the paper's arxiv version and code are available on 10/10/2021. The zero time-overlap prevents us to include a discussion with this new paper in our submission. Below, we are happy to provide comparisons and include them in the revised manuscript.

**(Similarity and Difference)** We acknowledge that the two works share a similar idea of utilizing weak labels of data in contrastive learning. [1] motivates from preventing class collision during instance-wise contrastive learning (random data that belongs to the same category will possibly get falsely pushed away in instance-wise contrastive learning), and ours motivates from exploring structural information of data within contrastive learning, followed by providing information-theoretic analysis to explain how different structural information can affect the learned representations. Task-wise, [1] focuses on unsupervised (no access to data labels) and semi-supervised (access to a few data labels) representation learning, and ours focuses on weakly-supervised (access to side information such as data attributes) and unsupervised representation learning. For the common unsupervised representation learning part, [1] presents to generate weak labels using connected components labeling process, and ours generates weak labels using K-means clustering.

**(Empirical Results)** We observe that the performance on ImageNet-100 reported in [1] looks better than ours (79.77 [1] v.s. ours 77.9). However, the experimental settings differ a lot. First, the *datasets* are different despite the same name: [1] considers ImageNet-100 by selecting the first 100 class of the ILSVRC 2012 challenge, and we select a different set of 100 classes (details shown in the Appendix). Second, the *batch size* is different: [1] considers 2048 , and ours considers 128. Third, the *projection heads in architecture* are different: [1] uses 2 projection heads (each with 4096 hidden units) with two objectives, one is for InfoNCE and the other is for the proposed Weakly Supervised Contrastive Learning loss; whereas ours uses one projection head with 2048 hidden units for Cl-InfoNCE objective only. Although our main experiments have demonstrated that Cl-InfoNCE alone can achieve competitive performance, we acknowledge that adding InfoNCE objective with an additional linear projection head would further improve the learned representation. To fairly compare our Cl-InfoNCE loss with their proposed Weakly Supervised Contrastive objective, we add an additional projection head that is trained with InfoNCE along with our Cl-InfoNCE objective. Experiments are conducted on our version of ImageNet100 with the controlled setup: same network architecture of resnet50, same projection head design (two heads of 2048-2048-128), same batch size of 384, same training epochs of 200, and the same optimizer and linear evaluation protocols (top-1 accuracy is reported). The results are shown below.


| Method | Linear Evaluation (%) |
| -------- | -------- |
| ICCV 2021 [1]    | 82.2 +- 0.4 |
| Cl-InfoNCE + Kmeans (ours) |  82.6 +- 0.2 |

From the results, we can see that the two methods' performances are similar. Our work and theirs [1] are done independently and concurrently, and both works allow a broader understanding of weakly supervised contrastive learning. Please let us know if there are any more concerns.

[1] Weakly Supervised Contrastive Learning. ICCV 2021.

---

> ### Comment · Area_Chair_9bty · 2021-11-30
> **difference in ImageNet class selection**
>
> Hi authors, thanks for the detailed response. I'm a bit confused, even after reading appendix E, about how you selected 100 classes from ImageNet and what was the rationale behind it. It would be better for the community to have a standardized smaller subset, instead of different sets named ImageNet-100 used for evaluation.

---

> > ### Author Response · Authors · 2021-11-30
> > **Follow-up**
> >
> > Dear AC,
> >
> > Thank you for the comment and suggestion! We select 100 classes from ImageNet to simplify the hierarchy cluster construction used in Appendix B. In Appendix B, we use WordNet annotation to construct hierarchy clusters (i.e., if two classes have the same parent in WordNet hierarchy, they belong to the same cluster). However, this construction requires classes to only have a single path connected to the root in WordNet. Since "ImageNet-100" used in [1] does not satisfy this requirement, we construct our own. We are sorry for the naming, and we will change the name of our imagenet-100 version to "ImageNet-100*", with a star attached at the end.
> >
> > To address your concern, we will perform experiments on the "ImageNet-100" used in [1] and report the numbers. It takes approximately three days, and we will include the results once we have the numbers. If the forum is closed then, we will include the results in our anonymous github link: https://anonymous.4open.science/r/Cl-InfoNCE-02AB
> > We will also include these experiments in the final manuscript; as of now, we are not able to update the manuscript.

---

> > ### Author Response · Authors · 2021-12-04
> > **Follow-up Part II**
> >
> > Dear AC,
> >
> > We performed the following experiments. We ran the ImageNet version provided in ICCV work [1] and reported our and theirs results [1]. For our result, we report the number using the batch size $128$. We directly run their code for their [1] result and change the batch size to $128$. The only difference between our method and their method [1] is the algorithm, fixing the model and optimizer.
> >
> > Method           | Ours  | [1] |
> > --------------|-------|------|
> > Top-1 Accuracy    | $74.04$ |  $73.68$ |
> >
> > The new result will be provided in the final manuscript with clear notice about the difference in ImageNet-100 versions. Please let us know if you need more clarifications or experiments.
> >
> > [1] Weakly Supervised Contrastive Learning. ICCV 2021.

---

### Comment · Area_Chair_9bty · 2021-11-24
**Discussion period**

Dear reviewers,

The authors have provided us with detailed response, as well as updated paper draft.
Could you take a look and comment on how this changes your initial evaluation?
Even when your evaluation does not change, it will be helpful to know why.

best,
Area Chair

---

### Decision · Program_Chairs · 2022-01-20

**Decision:**

Accept (Poster)

**Comment:**

The paper proposes a weakly supervised contrastive learning, using auxiliary cluster information, for representation learning. Their method generates similar representations for the intra-cluster samples and dissimilar representations for inter-cluster samples via a clustering InfoNCE objective. Their approach is evaluated thoroughly on three image classification task.

The reviewers agree that the paper is well written, presenting interesting theoretical analysis (Reviewer h3zd,  a8kw) and solid experimetal results (Reviewer RhYi, 1ziy). The core idea of the paper is relatively simple and well motivated (Reviewer h3zd). While the focus is using the clustering with auxiliary labels, the method can be applied without auxiliary labels with K-means.
There were some concerns from the reviewers:  the overlap with a concurrent work [1]. The authors have provided detailed discussions on conceptual (concurrent work focuses on unsupervised cases where this work focuses on weakly-supervised setting) and emprical comparisons. Accordingly, reviewer a8kw and 1ziy had some issues with the novelty of the paper, as it can be interpreted as slight modification from previously explored idea (vanilla InfoNCE loss).

Despite some overlap with existing approaches, the paper presents an interesting and well conducted study of integrating clustering information for learning representation, so I vote for acceptance.

[1] Weakly Supervised Contrastive Learning. ICCV 2021.